# Accessing Higher Dimensions for Unsupervised Word Translation

**Sida I. Wang**
FAIR, Meta

## Abstract

The striking ability of unsupervised word translation has been demonstrated recently with the help of low-dimensional word vectors / pretraining, which is used by all successful methods and assumed to be necessary. We test and challenge this assumption by developing a method that can also make use of high dimensional signal. Freed from the limits of low dimensions, we show that relying on low-dimensional vectors and their incidental properties miss out on better denoising methods and signals in high dimensions, thus stunting the potential of the data. Our results show that unsupervised translation can be achieved more easily and robustly than previously thought – less than 80MB and minutes of CPU time is required to achieve over 50% accuracy for English to Finnish, Hungarian, and Chinese translations when trained in the same domain; even under domain mismatch, the method still works fully unsupervised on English NewsCrawl to Chinese Wikipedia and English Europarl to Spanish Wikipedia, among others. These results challenge prevailing assumptions on the necessity and superiority of low-dimensional vectors and show that the higher dimension signal can be used rather than thrown away.

## 1 Introduction

The ability to translate words from one language to another without any parallel data nor supervision has been demonstrated in recent works (Lample et al., 2018b; Artetxe et al., 2018b, ...), has long been attempted (Rapp, 1995; Ravi and Knight, 2011, ...), and provides empirical answers to scientific questions about language grounding (Bender and Koller, 2020; Søgaard, 2023). However, this striking phenomenon has only been demonstrated with the help of pretrained word vectors or transformer models recently, lending further support to the necessity and superiority of low-dimensional representations. In this work, we develop and test *coocmap*[1] for unsupervised word translation using only simple co-occurrence statistics easily computed from raw data. coocmap is dual method of vecmap (Artetxe et al., 2018b), using co-occurrences statistics in place of low-dimensional vectors. The greedy and deterministic coocmap establishes the most direct route from raw data to the striking phenomenon of unsupervised word translation, and shows that pretrained representation is not the key.

More surprisingly, coocmap greatly improves the data efficiency and robustness over the baseline of vecmap-fasttext, showing that relying on low-dimensional vectors is not only unnecessary but also inferior. With coocmap, 10–40MB of text data and a few minutes of CPU time is sufficient to achieve unsupervised word translation if the training corpora are in the same domain (e.g. both on Wikipedia, Figure 1). For context, this is less than the data used by Brown et al. (1993) to train IBM word alignment models. On cases that were reported not to work using unsupervised methods by Søgaard et al. (2018), we confirm their findings that using fasttext (Bojanowski et al., 2017) vectors indeed fails, while coocmap solved many of these cases with very little data as well. For our main results, we show that less similar language pairs such as English to Hungarian, Finnish and Chinese posed no difficulty and also start to work with 10MB of data for Hungarian, 30MB of data for Finnish

---

[1]code released at `https://github.com/facebookresearch/coocmap`

37th Conference on Neural Information Processing Systems (NeurIPS 2023).

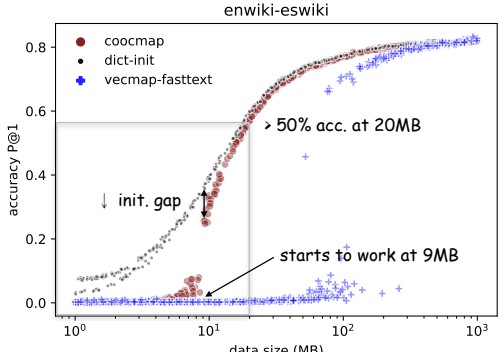

Figure 1: Our results focus on the data requirements as we varying the amount and domains of data. The unsupervised accuracy increases quickly with a sharp starting point point and this can happen surprisingly early (~10MB here), reaching >50% by 20MB. As data increases, supervised initialization with ground-truth dictionary (dict-init) does not make much difference as shown by the vanishing initialization gap. vecmap-fasttext needs more data and is less stable.

and 20MB of data for Chinese. The hardest case of training with domain mismatch (e.g. Wikipedia vs. NewsCrawl or Europarl) is generally reported not to work (Marchisio et al., 2020) especially for dissimilar languages. After aggressively discarding some information using clip and drop, coocmap works on English to Hungarian and Chinese, significantly outperforming vectors, which do not work.

These results challenge the conventional wisdom ever since word vectors proved their worth both for cross-lingual applications (Ruder et al., 2019) and more generally where a popular textbook claims *dense vectors work better in every NLP task than sparse vectors* (Jurafsky and Martin, 2023, 6.8). We discuss reasons that disadvantage low-dimensional vectors unless compression is the goal, which is far more intuitive than the conventional wisdom that *dense vectors are both smaller and perform better*. For vectors to perform, they must have nice linear-algebraic properties (Mikolov et al., 2013c,b), successfully denoise the data, while still retaining enough useful information. The linear algebraic and denoising properties are *incidental*, since they are not part of the training objective but are a consequence of being in low dimensions. These incidental properties come in conflict with the actual training objective to retain more information as dimension increases, leaving a small window of opportunity for success. The incidental denoising we get from having low dimensions, while interesting and sufficient in easy cases, is actually suboptimal and worse than the more intentional denoising in high dimensions used by coocmap. Furthermore, without the need to have low dimensions, coocmap can access useful information in higher dimensions. The higher dimensions contain knowledge such as *portland-oregon, cbs-60 minutes, molecular-weight, luisiana-purchase, tokyo-1964* that tend to be lost in lower dimensions. We speculate that similarly processed co-occurrences would outperform low-dimensional vectors in other tasks too, especially if the natural but incidental robustness of vectors is not enough.

## 2 Problem formulation

This word translation task is also called lexicon or dictionary induction in the literature. The task is to translate words in one language to another (e.g. *hello* to *bonjour* in French) and is evaluated for accuracy (precision@1) on translations of specific words in a reference dictionary. We do this fully unsupervised, meaning we do not use any seed dictionary or character information. Let the datasets $D_1$ and $D_2$ be sequences of words from the vocabulary sets $V_1, V_2$. Since we do not use character information, we may refer to vocabulary by their integer indices $V = [1, 2, \ldots]$ for convenience. We would like to find mapping from $V_1$ to $V_2$ based on statistics of the datasets. In particular, we consider the window model where the sequential data is reduced to pairwise co-occurrences over context windows of a fixed size $m$. The word-context co-occurrence matrix $\text{Co} \in \mathbb{R}^{|V| \times |V|}$ counts the number of times word $w$ occurs in the context of $c$, over a window of some size $m$

$$\text{Co}(w, c) = \sum_{i=1}^{|D|} \sum_{-m \leq j \leq m, j \neq 0} \mathbb{I}[w_i = w, w_{i+j} = c] \tag{1}$$

where $w_i$ is the $i$-th word of the dataset $D$. The matrix $\text{Co}$ is the sufficient statistics of the popular word2vec (Mikolov et al., 2013a,c) and fasttext (Bojanowski et al., 2017), which use the same

co-occurrence information, including additional objectives not explicit in the loss function:

$$\ell(\theta) = \sum_{i=1}^{|D|} \sum_{-m \leq j \leq m, j \neq 0} \log_\theta p(w_{i+j}|w_i). \tag{2}$$

For word translation, we obtain $\mathrm{Co}_1$ and $\mathrm{Co}_2$ from the two languages $D_1$ and $D_2$ separately.

**Multilingual distributional hypothesis.** If words are characterized by its co-occurrents (Harris, 1954), then translated words will keep their translated co-occurrents. In more precise notation, let $(s_1, t_1), (s_2, t_2), \ldots, (s_n, t_n)$ be $n$ pairs of translated words, then for translation $(s, t)$

$$X[s, s_1], X[s, s_2], \ldots, X[s, s_n] \sim Z[t, t_1], Z[t, t_2], \ldots, Z[t, t_n],$$

for association matrices $X = K(\mathrm{Co}_1)$, $Z = K(\mathrm{Co}_2)$. While intuitive, this is not known to work unsupervised nor with minimal supervision. Rapp (1995) presents evidence and correctly speculates that there may be sufficient signal. Fung (1997); Rapp (1999) used a initial dictionary to successfully expand the vocabulary further using a mutual information based association matrix. Alvarez-Melis and Jaakkola (2018) operates on an association matrix generated from word vectors.

**Isomorphism of word vectors.** Despite the clear motivation in the association space, unsupervised translation was first shown to work in vector space, where a linear mapping between the vector spaces corresponds to word translation. This is called (approximate) *isomorphism* (Ruder et al., 2019). If $s$ translate to $t$, and $X_s, Z_t$ are their vectors, then $X_s W \sim Z_t$ where $W$ can be a rotation matrix, and the association metric can be cosine-distance, often after applying normalizations to raw word vectors. Successful methods solve this problem using adversarial learning (Lample et al., 2018b) or heuristic initialization and iterative refinement (Artetxe et al., 2018b; Hoshen and Wolf, 2018), among others (Zhang et al., 2017b).

## 3 Method

We aim for simplicity in coocmap, and all steps consists of discrete optimization by arranging columns of $X$ or measuring distances between rows of $X$ where $X$ is the association matrix based on co-occurrences (and $Z$ for the other language). Almost all operations of coocmap has an analog in vecmap and we will describe both of them for clarity. There are two main steps in both methods, finding best matches and measuring cdist pairwise distances. Each row of $X$ corresponds to a word for both vecmap and coocmap and are inputs to cdist. vecmap finds a rotation $W$ that best match given row vectors $X[s, :]$ and $Z[t, :]$ for $s = [s_1, s_2, \ldots, s_n] \in V_1^n$ and $t = [t_1, t_2, \ldots, t_n] \in V_2^n$. For coocmap, instead of solving for $W$, we just re-arrange the columns of the association matrix with indices $s, t$ directly to get $X[:, s], Z[:, t]$ where the columns of $X$ are words too.[2]

---

| **Algorithm 1** coocmap self-learning | **Algorithm 2** vecmap self-learning |
|---|---|
| $X \in \mathbb{R}^{|V_1| \times |V_1|}, Z \in \mathbb{R}^{|V_2| \times |V_2|}$; | $X \in \mathbb{R}^{|V_1| \times d}, Z \in \mathbb{R}^{|V_2| \times d}$; |
| **Input** $s, t$    **Output** $s, t$ | **Input** $s, t$    **Output** $s, t$ |
| **repeat** | **repeat** |
| | $\quad W = \mathrm{solve}(X[s, :], Z[t, :])$ |
| $\quad D = \mathrm{cdist}(X[:, s], Z[:, t])$ | $\quad D = \mathrm{cdist}(XW, Z)$ |
| $\quad s, t = \mathrm{match}(D)$ | $\quad s, t = \mathrm{match}(D)$ |
| **until** no more improvement | **until** no more improvement |

---

$\mathrm{cdist}(X, Z)_{ij} = \mathrm{dist}(X_i, Z_j)$ is a function that takes pairwise distances for each row of both inputs and then outputs the results in a matrix. The self-learning loop use the same improvement measurement as in vecmap $\mathrm{mean}_i \max_j(\mathrm{cdist}(i, j))$, but without stochastic search. We explain how to generate $X, Z, s, t$, and match next.

**Measurement.** It is important to normalize before taking measurements, and vecmap normalization consists of normalizing each row to have unit $\ell_2$-norm (unitr), center so each column has 0 mean (centerc), and normalizing rows (unitr) again. For precision,

$$\mathrm{normalize}(Y) := \mathrm{unitr}(\mathrm{centerc}(\mathrm{unitr}(Y))), \tag{3}$$

where $\mathrm{centerc}(Y) := Y - \mathrm{sumr}(Y)/r$, $\mathrm{sumr}(Y) := \mathbf{1}^T Y$, and $Y \in \mathbb{R}^{r \times c}$.

---

[2]We show a derivation in E on the equivalence of vecmap and coocmap.

**coocmap.** The input matrix $X = \text{normalize}(\text{Co}^{\circ\frac{1}{2}})$ is obtained from Co by taking elementwise square-root then normalize. cdist is the cosine-distance.

**vecmap.** From original word vectors $X'$, we use their normalized versions $X = \text{normalize}(X')$. For $\text{solve}(X[s], Z[t])$, we use $\arg\min_{W \in \Omega} ||X[s,:]W - Z[t,:]||_F$, for rotation matrices $\Omega$. This is known as the Procrustes problem. cdist is the cosine-distance.

**Initialization.** We need the initial input $s, t$ for Algorithms 1 and 2. For the unsupervised initialization, we follow vecmap's method based on row-wise sorting. Let $\text{sort\_row}(X)$ make each row of $X$ be sorted independently and $\text{normalize}(X)$ is defined in (3), then

---

**Algorithm 3** unsupervised initialization

---
**Input** $X, Z$,   **Output** $D$
$R(Y) := \text{sort\_row}(K(Y))$
$S(Y) := \text{normalize}(R(Y))$
$D = \text{cdist}(S(X), S(Z))$

---

vecmap: $X, Z \in \mathbb{R}^{|V| \times |V|}$
$X = (X'X'^\mathsf{T})^{\frac{1}{2}}$ for $X' \in \mathbb{R}^{|V| \times d}$

coocmap: $X, Z \in \mathbb{R}^{|V| \times |V|}$
$X = \text{normalize}(\text{Co}_1^{\circ 1/2})$

The first step of vecmap $X = (Y'Y'^\mathsf{T})^{\frac{1}{2}}$ is actually converting from vector space to the association space. So it is natural to replace the first step by $\text{normalize}(\text{Co}_1^{\circ 1/2})$ for coocmap.

**Matching.** The main problem once we have a distance matrix is to solve a matching problem

$$\min_M \sum_{(i,j) \in M} \text{cdist}(i,j) \tag{4}$$

vecmap proposes a simple matching method, where we take $j^* = \arg\min_j \text{cdist}(i,j)$ for each $i$ in forward matching, and then take $i^* = \arg\min_i \text{cdist}(i,j)$ for each $j$ in backward matching. This always results in $|V_1| + |V_2|$ matches where words on each side is guaranteed to appear at least once. For coocmap, there is complete freedom in forming matches $i, j$ and often many words all match with a single word. As a result, hubness mitigation (Lazaridou et al., 2015) is even more essential compared to vecmap.

While there are many reasonable matching algorithms, we find that Cross-Domain Similarity Local Scaling (CSLS) (Lample et al., 2018b) was enough to stabilize coocmap. Note that CSLS acts on the pairwise distances and therefore directly applies to cdist,

$$\text{csls}(\text{cdist}(i,j)) = \text{cdist}(i,j) - \frac{1}{2k}\left(\sum_{j' \in N_i(k)} \text{cdist}(i,j') + \sum_{i' \in N_j(k)} \text{cdist}(i',j)\right)$$

where $N_i(k)$ are the $k$-nearest neighbors to $i$ according to cdist. We use $\text{csls}(\text{cdist})$ as the input to (4) in place of cdist. Instead of always preferring the best absolute match, CSLS prefers matches that stand out relative to the $k$ best alternatives.

### 3.1 Clip and drop

This basic coocmap above already works well on the easier cases and these techniques provided small improvements. However, for difficult cases of training on different domains, clip is essential and drop gives a final boost that can be large. These operations are aggressive and throws away potentially useful information just like rank reduction, but we will show that they are far better especially under domain shifts.

**clip.** For clip, we threshold the top and bottom values by percentile. While there are just 2 numbers for the 2 thresholds, we use two percentiles operations to determine each of them so they are not dominated by any particular row. Let $r_i = Q_p(X_i)$ be the $p$th percentile value of the row $i$ of $X$. We threshold $X$ at $Q_p(r_1, r_2, \ldots, r_{|V|})$, where $p = 1\%$ for lowerbound and $p = 99\%$ for upperbound. This results in two thresholds for the entire matrix that seem to agree well across languages. For results on clip, we run coocmap with $X = \text{clip}(\text{normalize}(\text{Co}^{\circ 1/2}))$. Intuitively, if words are already very strongly associated, obsessing over exactly how strongly is not robust across languages and datasets but can incur large $\ell_2$s. For lowerbound, extremely negatively associated words is clearly not robust, since the smallest count is 0 and one can always use any two words together. Both bounds seem to improve over baseline, but the upperbound is the more important one.

**drop (head).** Drop the $r = 20$ *largest* singular vectors, i.e. $\text{drop}_r(X) = X - X_r \in \mathbb{R}^{|V| \times |V|}$, where $X_r$ is the rank-$r$ approximation of $X$ by SVD. In the experiments, we first get the solution of $s, t$ from $\text{clip}(\text{normalize}(\text{Co}^{\circ 1/2}))$ then use coocmap on $X = \text{clip}(\text{drop}_r(\text{normalize}(\text{Co}^{\circ 1/2})))$ with $s, t$ as the initial input. Drop was good at getting a final boost from a good solution, but is usually worse than the basic coocmap in obtaining an initial solution. While the dropping top 1 singular vector is discussed by (Mu and Viswanath, 2018; Arora et al., 2017) and popular vectors implicitly drop already (see A), we see more benefits from dropping more and this seems to enable more benefits of the higher dimensions. We show examples in Appendix F in support of this viewpoint.

**Truncate (tail).** This is the usual rank reduction where $\text{trunc}_r(X) = X_r$ is the best rank-$r$ approximation of $X$ by SVD. We use this for analysis on the effect of rank.

## 4 Experiments

For the main results, we report accuracy as a function of data size and only show results in the fully unsupervised setting. The accuracy is measured by precision at 1 on the full MUSE dictionary for the given language pair on the 5000 most common words in each language (see Section 6 for limitations). Many results will be shown as scatter plots of accuracy vs. data size, each containing thousands of experiments and more informative than tables. They will capture stability and the qualitative behaviors of the transitional region as the amount of data varies. Each point in the scatter plots represents an experiment where a specific amount of data was taken from the head of the file for training co-occurrence matrices and fasttext vectors with default settings (skipgram, 300 dimension, more in B) for fasttext. coocmap use the same window size as fasttext ($m = 5$), the same CSLS ($k = 10$) and same optimization parameters as vecmap. coocmap does not require additional hyperparameters until we add clip and drop. The same amount of data is taken from both sides unless one side is exhausted. In our experiments, most cases either achieve > 50% accuracy (i.e. *works*) or near 0 accuracy and has a definite starting point (i.e. *starts to work*). Summary of these results are in Table 1 and we will show details on progressively harder tests.

**Methods.** we compare these methods for the main results.

- **dict-init**: initialize with the ground truth dictionary then apply coocmap.
- **coocmap**: fully unsupervised coocmap and improvements with **-clip**, **-drop** if needed.
- **vecmap-fasttext**: apply vecmap to 300 dimensional fasttext vectors trained with default settings.
- **vecmap-raw**: apply vecmap to a svd factorization of the co-occurence matrix. If $\text{Co}^{\circ 1/2} = USV'$, then we use $US_r$ as the word vectors where $S_r$ only keeps top $r$ singular values. $r = 300$.

For dict-init, we initialized using the ground truth dictionary (i.e. initial input $s, t$ to Algorithm 1 are the true evaluation dictionary) and then test on the same dictionary after self-learning. This establishes an upper-bound for the amount of gains possible with a better initialization as long as we are using the basic coocmap measurements and self-learning. After coocmap works stably, their performance coincides, showing very few search failures and the limit of gains from initialization.

We use default parameters for fasttext in this section. This may be unfair to fasttext but better match other results reported in the literature where the default hyperparameters are used or pretrained vectors are downloaded. In Section B, we push on the potential of fasttext more and describe hyperparameters.

**Data.** We test on these languages paired with English (**en**): Spanish (**es**), French (**fr**), German (**de**), Hungarian (**hu**), Finnish (**fi**) and Chinese (**zh**).

For training data we use Wikipedia (**wiki**), Europarl (**parl**), and NewsCrawl (**news**), processed from the source and removing obvious markups so raw text remains. The data is processed using Huggingface WordLevel tokenizer with whitespace pre-tokenizer and lower-cased first.

**wiki** (`https://dumps.wikimedia.org/`): Wikipedia downloaded directly from the official dumps (pages-meta-current), extract text using WikiExtractor (Attardi, 2015) and removed `<doc id` tags. We start from the first dump until is >1000MB of data for each language and shuffle the combined file. For zh, we also strip away all Latin characters `[a-zA-Z]` and segment using jieba: https://github.com/fxsjy/jieba.

**parl** (`https://www.statmt.org/europarl/`): Europarl (Koehn, 2005) was shuffled after downloading since this is parallel data.

**news** (`https://data.statmt.org/news-crawl/`): NewsCrawl 2019.es was downloaded and used as is. For 2018-2022.hu and 2018.en, we striped meta data by `grep -v` on `http`, `trackingCode` and `{` after which a small random sample does not obviously contain more meta data. This removed over 35% from hu news and 0.1% from en.

For evaluation data, we use the *full* MUSE dictionaries for these language pairs. This allows for a more stable evaluation given our focus on qualitative behaviors, but at a cost of being less comparable to previous results.

| source | target | start | start' | works |
|--------|--------|-------|--------|-------|
| enwiki | eswiki | 9 | 70 | 20 |
| enwiki | frwiki | 10 | 80 | 30 |
| enwiki | zhwiki | 14 | 700 | 50 |
| enwiki | dewiki | 20 | 200 | 70 |
| enparl | huparl | 8 | – | 20 |
| enparl | fiparl | 30 | – | 80 |
| **Domain mismatch** | | | | |
| source | target | start | start' | works |
| enwiki | esnews | 30 | – | 70 |
| enwiki | hunews | 140 | – | 600 |
| enwiki | esparl | 500 | – | 500 |
| ennews | zhwiki | 800 | – | 800 |
| enwiki | huparl | – | – | – |
| enwiki | fiparl | – | – | – |

Table 1: Summary of data requirements for coocmap in **MB** (1e6 bytes). **start**: when coocmap starts to work, **start'**: when vecmap-fasttext baseline starts to work (check Figure 2 to see that **start** is clear), – denotes failure for the whole range; **works**: when coocmap reaches 50% accuracy. Readings are rounded up to the next tick of the log plot, or 1.4 if less than the middle of 1 and 2. The same amount of data is used on both source and target sides, unless one side is used up (e.g. 100MB of huparl or 300MB of esparl). There are less than 0.2 **million tokens per MB** in all datasets, ranging from 0.13 in fi-parl, 0.19 in zhwiki and 0.20 for ennews.

## 4.1 Same domain of data

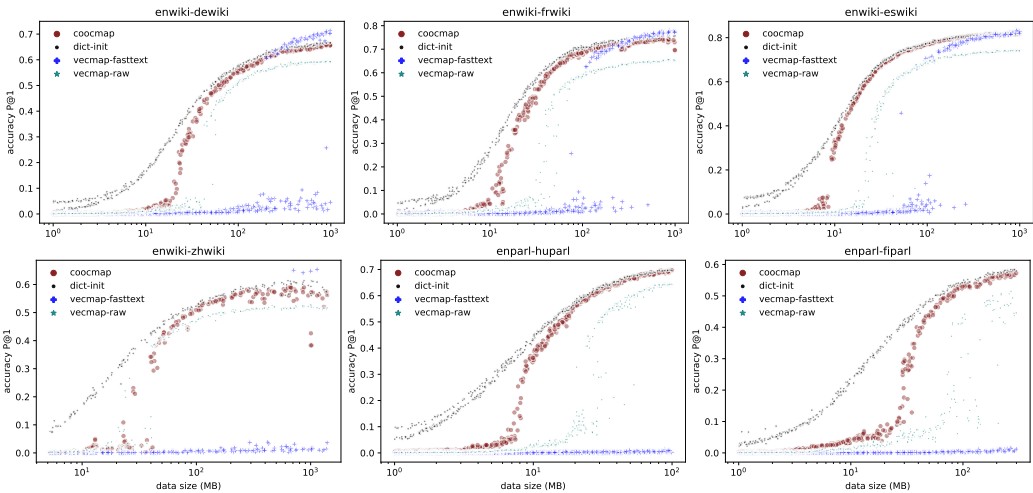

Figure 2: accuracy vs. datasize where the source data and target data are in the same domain, either both Wikipedia, or both Europarl (enparl-huparl, and enparl-fiparl). These cases are easy for coocmap but sometimes failed for vecmap-fasttext or required more data.

**Similar languages.** To establish some baselines, we start with the easy settings where everything eventually works. Figure 2 shows that coocmap starts to work at around 10MB of data while vecmap trained on fasttext only starts to work once there are over 100MB of data. The transitional region is fairly sharp, and coocmap works reliably on each language pair with around 100MB of data, and vecmap with fastText also mostly works once there is around 100MB of data. vecmap-fasttext eventually outperforms coocmap after it was trained on >100MB of data.

**Less similar languages.** For hard cases from (Søgaard et al., 2018), all of which were reported not to work using MUSE (Lample et al., 2018b) using fastText vectors. Here, we confirm that vecmap and fastText vectors also do not work on default settings. However, these are not even challenging to the basic coocmap where all cases started to work with less than 30MB of data.

For the small but clean Europarl data, we tested on English to Finnish (fi) and Hungarian (hu). As shown in Figure 2, coocmap started to work at 9MB for hu and 30MB for fi. It finished the transition region when we used all 300MB of en-fi and 100MB of en-hu. vecmap-fasttext has yet to work, agreeing with the results of Søgaard et al. (2018). However, since vecmap-raw using simple SVD worked, it was surprising that fasttext did not. Indeed, decreasing the dimension to 100 from 300 enables vecmap-fasttext to start working at 50MB of data for enparl-huparl (vs 8MB for coocmap, Figure 6).

Next result is on English to Chinese (zh), where both are trained on Wikipedia. coocmap had a bit more instability vecmap-fasttext also sometimes works with around 1GB of data. The more robust version of coocmap-clip and drop is completely stable, and begin to work with less than 20MB of data.

## 4.2    Under domain mismatch.

The most difficult case for unsupervised word translation is when data from different domains are used for each language (Søgaard et al., 2018; Marchisio et al., 2020). The original vecmap of Artetxe et al. (2018b) was tested on different types of crawls (WacKy, NewsCrawl, Common Crawl) which did not work in previous methods. Marchisio et al. (2022b) show that vecmap also failed for NewsCrawl to Common Crawl on harder languages. We test on Wikipedia to NewsCrawl and Europarl and see successes on enwiki-esnews, enwiki-hunews, and ennews-zhwiki, **enparl**-eswiki before finally failing on enwiki-fiparl and enwiki-huparl. See Figure 3.

On **enwiki-esnews**, where ~100MB of data was required to reach 50%, though the basic coocmap becomes unstable and all vecmap or fasttext based methods failed to work at all. However, clip and drop not only stablizes this but enabled it start to work at 40MB of data.

On **enwiki-hunews**, coocmap mostly fails but get 5% accuracy with above 100MB of data. Impressively, clip and drop fully solves this problem as well, but even clip has a bit of instability, reaching 50% accuracy at 600MB of data.

On **en*news*-zhwiki**, coocmap fails completely without clip and drop, and even then requires more data than before. Finally it works and reaching over 50% accuracy around 800MB. In C, we show the data still has higher potential, where truncating to 2000 dimensions enables ennews-zhwiki to work with 300MB or even 100MB of data, though still not reliably.

For more extreme domain mismatch, we test **enparl-eswiki**. In addition to being small, Europarl contains only parliamentary proceedings which has a distinct style and limited range of topics, to our surprise this also worked with 295MB of enparl, and 500MB from eswiki, reaching a final accuracy of 70% suddenly, although basic coocmap also showed no sign of working. All methods failed for enwiki-fiparl and enwiki-huparl in the range limited by Europarl (300MB for fiparl and 90MB for huparl) with up to 1GB of enwiki, reaching the end of our testing.

## 5    Analysis: why coocmap outperformed dense vectors

These main results show that high-dimensional coocmap is more data efficient and significantly more robust than the popular low-dimensional word vectors fasttext/word2vec, which contract prevailing assumptions that vectors are superior and necessary to enable unsupervised translation among other tasks. Ruder et al. (2019) states "word embeddings enables" various interesting cross-lingual phenomena. For unsupervised dictionary induction, landmark papers (Lample et al., 2018a; Artetxe et al., 2018b) needed vectors and even methods that *must* use a $|V| \times |V|$ input constructed these from low dimensional vectors anyway (Alvarez-Melis and Jaakkola, 2018; Marchisio et al., 2022a). More generally, the popular textbook Jurafsky and Martin (2023, 6.8) states "dense vectors work better in every NLP task than sparse vectors". Here, we provide natural reasons that disadvantage vectors if we do not mind having higher dimensions.

**The conflicting dimensions of vectors.**    These properties must hold for vectors to work,

1. *Approximate isomorphism*: be able to translate by rotation/linear map
2. *Denoise*: reduce noise and information not robust for the task
3. *Retention*: keep enough information at a given dimension to achieve good performance

By testing the dimension of word vectors in Figure 4, we can see that isomorphism and denoising only holds in low-dimensions. To see this on enwiki-dewiki where fasttext works well, both vecmap-

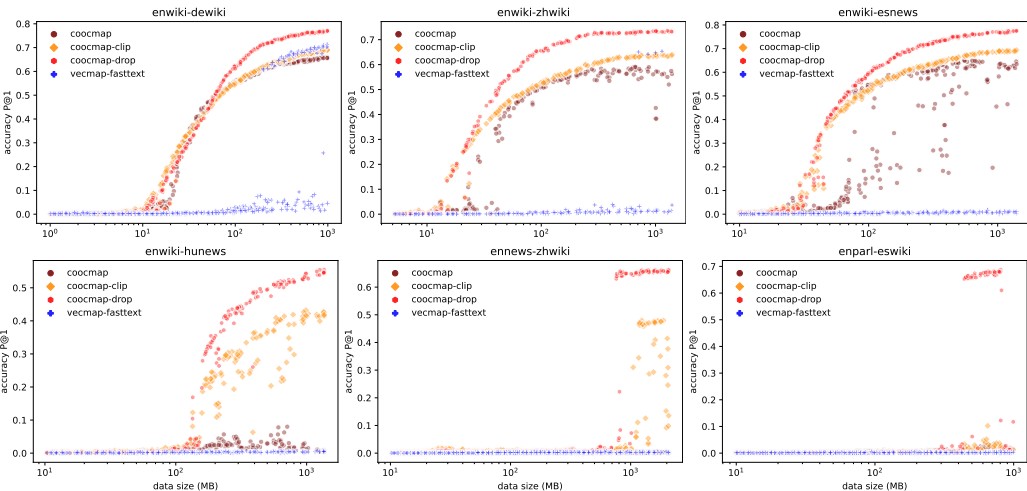

Figure 3: Accuracy vs. data size with clip and drop. Except for enwiki-eswiki and enwiki-zhwiki (top left, top middle), the rest all have domain mismatch where vecmap-fasttext gets ≈0.

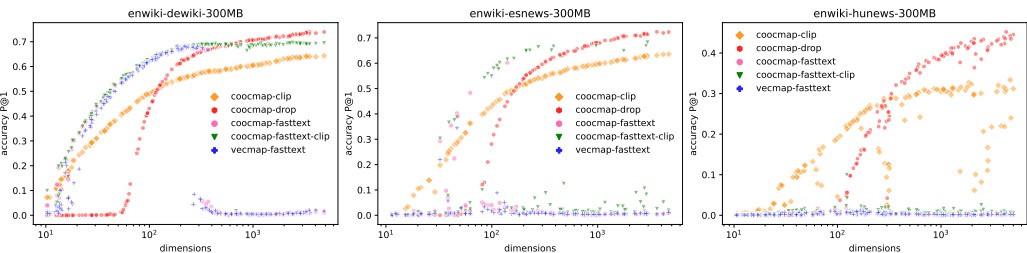

Figure 4: Accuracy vs. **dimension** at 300MB of data. coocmap-clip/drop: $\mathrm{trunc}_d(X)$ by svd, coocmap-fasttext: $X = \mathrm{normalize}((X'X'^{\mathsf{T}})^{\frac{1}{2}})$ for $X' \in \mathbb{R}^{|V| \times d}$, vecmap-fasttext: $X' \in \mathbb{R}^{|V| \times d}$.

fasttext and coocmap-fasttext are better than SVD truncation at low dimensions. Then accuracy goes to 0 as dimension increases. This is not because vectors fails to retain information, since coocmap-fasttext-clip works if we apply additional clipping. See B for more testing on the effect of dimension and other hyperparameters. Notably, lower dimensions have better isomorphism and denoising, and higher dimensions have better accuracy and retention. This trade off leaves a limited window for fasttext to work well. Still, selecting a good dimension is sufficient for fasttext vectors to match the accuracy and data efficiency of coocmap on the easiest cases (en-es, fr, de) and work well enough on dissimilar languages training on the same domain.

Søgaard et al. (2018) notes similar impact of dimensionality themselves, but without exploring enough range that would have solved their enparl-huparl and enparl-fiparl tests (with vecmap). Yin and Shen (2018) argues that optimal dimension may exist because of bias-variance tradeoff. Our experiments show that coocmap keeps improving with higher dimensions, and they may have been misled by relying on linear algebraic properties. The unreliability of isomorphism is also noted by Patra et al. (2019); Marchisio et al. (2022b).

**Better denoising in high dimensions.** Domain mismatch is where the vectors struggle the most regardless of dimension. In the easiest domain mismatch case of enwiki-esnews of Figure 4, vecmap-fasttext failed whereas coocmap-fasttext-clip worked (though not stably), showing that **clip** helps even when applied on top of the natural but incidental denoising of vectors.

The association matrix of coocmap $\mathrm{normalize}(\mathrm{Co}^{\circ 1/2})$ is also better overall than other choices such as Mikolov et al. (2013a); Levy and Goldberg (2014); Pennington et al. (2014); Rapp (1995). In A, we compared to other full-rank association matrices corresponding to popular vectors and show they also perform worse than coocmap. For instance, the positive pointwise mutual information matrix (PPMI) of Levy and Goldberg (2014) corresponds to word2vec/fasttext. While it can work with

simple $\ell_2$ cdist without normalize and can reach higher accuracy than the basic coocmap if tested on the same domain, it has lower data efficiency and completely fails on the easiest enwiki-esnews with domain mismatch, showing the same behaviors as fasttext. A more aggressive squishing such as $\mathrm{normalize}(\log(1 + \mathrm{Co}))$ works robustly too but is less accurate and less data efficient. Impressively, Rapp (1995) also works fully unsupervised with the prescribed $\ell_1$.

**Higher dimensions contain useful information.** In all cases, coocmap-clip and coocmap-drop both become more accurate as dimension increases. Under domain mismatch, there is considerable gains above even 1000 dimensions, suggesting that we are losing useful information when going to lower dimensions. Analyzing the association matrices based on which values are clipped, we show in F that higher dimensions retain more world knowledge such as *portland-oregon, cbs-60 minutes, molecular-weight, basketball-jordan, luisiana-purchase, tokyo-1964* and using low dimensional vectors misses out.

## 6 Discussions

**Better vectors exist.** coocmap shows that it is feasible to denoise in high dimensions better than with popular vectors. Thus, applying denoising suitable to task, the size/type of training data first before reducing dimension is a more principled ways to train lower dimensional vectors than the current practice of relying on poorly optimizing a supposed objective. In this view, training vectors can just focus on how to best retain information in low dimensions and vectors should just get monotonously better with increasing dimensions and stop at an acceptable loss of information for the task and data.

There is room for improvements for vectors of all dimensions. In Figure 4, fasttext got higher accuracy at impressively low dimensions in favorable conditions whereas SVD on coocmap-drop becomes more accurate with increasing dimension. Without actually constructing the better vectors, it is clear that meaningful improvements are possible: *for any given dimension d, there exists rank-d matrices that can achieve a higher accuracy than the max of all rank-d methods already tested in Figure 4.*

**But why vectors?** Here is our opinion on why low-dimensional vectors appeared necessary and better beside social reasons. Co-occurrence counts from real data are noisy in very involved ways and violate standard statistical assumptions, however it is obligatory to apply principled statistics such as likelihood based on multinomial or $\chi^2$, whereas vectors take away these bad options and allow less principled but better methods. Statistics says likelihood ratios is the most data efficient: $D(s, t) = \sum_i p(s, i) \log \frac{p(s,i)}{p(t,i)}$, or perhaps the Hellinger distance $H(s, t) = \sum_i ||p(s, i)^{\frac{1}{2}} - p(t, i)^{\frac{1}{2}}||_2^2$ if we want more robustness. Using raw counts or a term like $p \log p$ (super-linear in raw counts) always failed when measuring with the $\ell_2$ norm. The likelihood-based method of Ravi and Knight (2011) likely would fail on real data for these reasons. Normalization is a common and crucial element of all working methods. The methods of Rapp (1995); Fung (1997); Rapp (1999) were based on statistical principles but with modifications to make normalized measurements. Levy and Goldberg (2014); Pennington et al. (2014) actually proposed better normalization methods too that would equally apply to co-occurrences (Appendix A).

Giving up on statistics and starting from vecmap lead to big improvements, ironic for our attempt to understand unsupervised translation and how exactly vectors enabled it. To prefer more empirical tests on a wider range of operations instead of believing that we can model natural language data with simple statistics might be another underrated lesson from deep learning. In this work, we give up on statistical modelling without giving up on higher dimensions, which makes available these key tools that were sufficient to outperform low-dimensional vectors: large squeezing operations like sqrt, normalizing against a baseline (subtracting means, divide by marginals), contrasting with others during matching (CSLS), clip and drop. With just normalize and CSLS, coocmap would have similar accuracy and higher data efficiency compared to fasttext. Clip made co-occurrences more robust than vectors while also increasing accuracy, showing the possibility of denoising in high-dimensions. Finally, drop enabled some access to the information in higher dimensions and lead to noticeably higher accuracy as dimension increases. This is actually a more intuitive situation than the prevailing understanding that going to very low dimensions is both more compact and more accurate.

**IBM models.** coocmap shows that unsupervised word translation would have been possible with the amount of compute and data used to train IBM word alignment models (Brown et al., 1993).

While we do not test on the exact en-fr Hansards (LDC), we know it has > 1.3 million sentence pairs from the Parliament of Canada. This can be compared to our Europarl en-hu experiments, which has 0.6 million sentence pairs for a more distant language pair, showing there may not be much additional information from the sentence alignments. coocmap with drop got around 75% accuracy here. Though the compute requirement of coocmap may have been difficult in '93 – by our estimate en-hu with $|V| = 5000$ probably requires $|V|^3 * 100 \approx 1$ Tera-FLOs, which should have been easy by the early 2000s on computers having Giga-FLOPS.

**Limitations.** The vocabulary size is a potentially important parameter unexplored in this paper where we used a low 5000 for cleaner/faster experiments. The most expensive step is factorization operations and measuring distances of high dimensional co-occurrences which is $O(|V|^3)$ as opposed to $O(d|V|^2)$ for vectors, though approximations are available with dimension reduction (see E). Having a low vocabulary size may explain why we see little benefit (except in our hardest cases such as C) from the potential denoising effects of truncation. Low-dimensional vectors is more easily scalable to larger vocabulary and can more easily used by neural models. However, BPE, indexing, truncating vocabulary, multiple window sizes and other alternatives are not explored to better use higher dimensions.

**Speculations.** As cross-domain is more challenging than just cross-lingual, it is especially likely that our findings may apply in other cross-domain tasks. So we speculate that if co-occurrences are similarly processed without going to low dimensions, they are likely to perform better on other tasks where more robustness is required than the natural robustness of low-dimensions. Similarly, neural models may also benefit from intentional denoising like clip and drop.

Beyond simple vectors, more recent language models may also suffer from low dimensions, which might be why they benefit from retrieval (Khandelwal et al., 2020), where low dimensional vectors may lose too much information for the task. Recently, contextual transformer representations superseded non-contextual vectors with more data and more dimensions pushing up the performance across tasks. Following Kaplan et al. (2020), there are important trade-off between dimensions, data and compute. If data is the limiting factor, it is plausible that better ways of using higher dimensions is helpful.

**Related work.** Rapp (1995) already hinted that there might be enough signal for fully unsupervised translation. Fung (1997); Rapp (1999); Haghighi et al. (2008) tested their high dimensional approaches with seed words, but without a full search procedure. Ravi and Knight (2011) succeeded in the case of time expressions and parallel data, but likely used insufficient normalization for harder cases. Mikolov et al. (2013b) noticed the relation between low-dimensional rotations and translation. With good enough vectors, Zhang et al. (2017a); Lample et al. (2018a); Artetxe et al. (2018a) achieved fully unsupervised word translation while showing that seed words do not matter too much as long as the unsupervised method works. There was much follow up on these successes, in particular, even methods that must use an association matrix (Alvarez-Melis and Jaakkola, 2018; Marchisio et al., 2022b) constructed them from low-dimensional vectors.

## Broader Impact

For dictionary induction, this work shows it can be done with less data and is more robust to domain mismatch than previously thought. We have a working procedure that is more direct, using less compute, making it more accessible for educational purposes. Unsupervised translation means learning from the statistics of data alone and can make bland and silly mistakes where outputs should not be relied on unless they can be verified.

## Acknowledgments

Thanks to Mikel Artetxe, Kelly Marchisio, Luke Zettlemoyer, Scott Yih, Freda Shi, Hila Gonen and Tianyi Zhang for helpful discussions.

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

# A  Comparison of association matrices

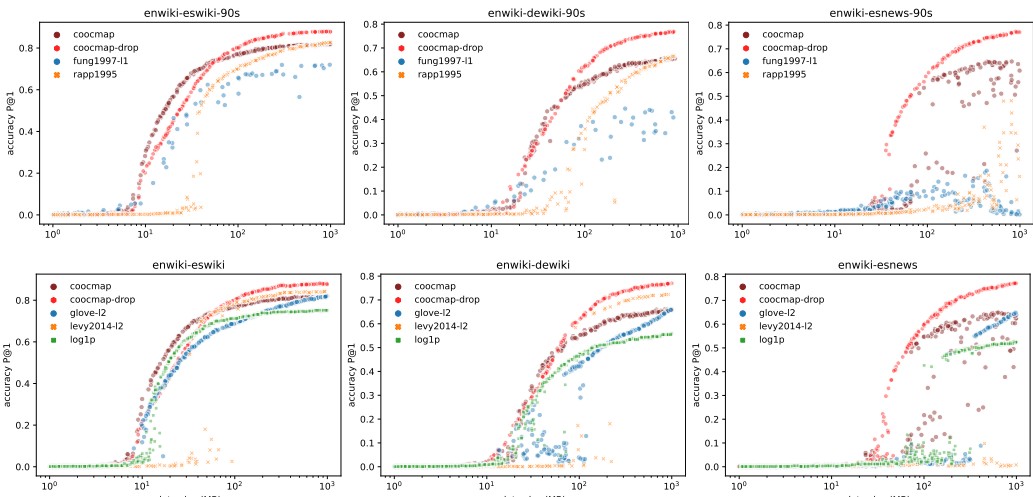

Figure 5:  *top*: methods from the 90s, *bottom*: methods from 2010s word2vec and glove.

| method | association formula | normalize | cdist |
|---|---|---|---|
| coocmap | $\mathrm{Co}^{\circ 1/2}$ | normalize | $\ell_2$ |
| log1p | $\log(1 + \mathrm{Co})$ | normalize | $\ell_2$ |
| Rapp (1995) | $\frac{p(s,i)}{p(s)p(i)}$ | $\ell_1$ | $\ell_1$ |
| Fung (1997) | $p(s,i) \log \frac{p(s,i)}{p(s)p(i)}$ | $\ell_1$ | $\ell_1$ |
| Levy and Goldberg (2014) / Church and Hanks (1990) | $\max\left(0, \log \frac{p(s,i)}{p(s)p(i)} - \log k\right)$ | $\ell_2$ | $\ell_2$ |
| Pennington et al. (2014) | $\log(1 + \mathrm{Co}) - b \cdot 1^{\mathsf{T}} - 1 \cdot \tilde{b}^{\mathsf{T}}$ | $\ell_2$ | $\ell_2$ |

In Figure 5, we compare to several choices of association method $K(\mathrm{Co})$ and find all of them to be worse than $\mathrm{Co}^{\circ 1/2}$ plus normalize overall, and the two popular methods from 2010s are better than the two methods from 1990s. These methods all perform normalization by factoring out the marginals one way or another.

Besides the association matrices themselves, we match them with their best normalization method from normalize, $\ell_1, \ell_2$. We report $\ell_2$ rather than normalize for Levy and Goldberg (2014); Pennington et al. (2014) which proposed their own normalization. In all these cases, normalize gave almost identical results as $\ell_2$, showing their own normalization was sufficient. normalize corresponds to $\ell_2$ cdist since it did $\ell_2$ normalization as the last step in (3). We use $p(s,i) = \mathrm{Co}(s,i)/\sum_{s',i'} \mathrm{Co}(s',i')$ when a probability was required.

On the top of Figure 5, we compare Rapp (1995); Fung (1997) from the 90s specifically designed for dictionary induction as well. Impressively, Rapp (1995) also performed well using $\ell_1$ distance as prescribed, where it even started to work on enwiki-esnews. Fung (1997) did not work with the prescribed $\ell_2$ criteria (not shown) but worked with $\ell_1$ as well. We think Rapp (1999) is similar to our modification of Fung (1997) with $\ell_1$, which contains its most important term. However, using $\ell_1$ lead to lower data efficiency and lower accuracy in methods that also work with $\ell_2$.

On bottom of Figure 5, we compare two popular association matrices from the 2010s. Notably, the positive PMI of (Levy and Goldberg, 2014) corresponds to word2vec/fasttext, and actually has similar performance characteristics as fasttext, where both reach higher accuracy than coocmap on easy cases but has lower data efficiency and fails on enwiki-esnews. Other methods that use normalize on log also worked for enwiki-esnews. Of these, coocmap has the highest accuracy. Levy and Goldberg (2014) achieved higher accuracy while Pennington et al. (2014) worked on enwiki-esnews. The normalization proposed in both methods achieved almost identical results with $\ell_2$ as with normalize. $\ell_1$ also worked on them but clearly worse and has less data efficiency (not shown).

Pennington et al. (2014) did not specify what the biases $b$ should be, instead just leaving them to gradient optimization, we picked them to subtract out the mean of $\log$s whereas PMI would subtract the $\log$ of means. $\text{Co}^{\circ 1/2}$ was also tested by Stratos et al. (2015) and found to be favourable. We also did not find the shifted PPMI to be better than the basic PPMI of Church and Hanks (1990).

## B   Improving fasttext

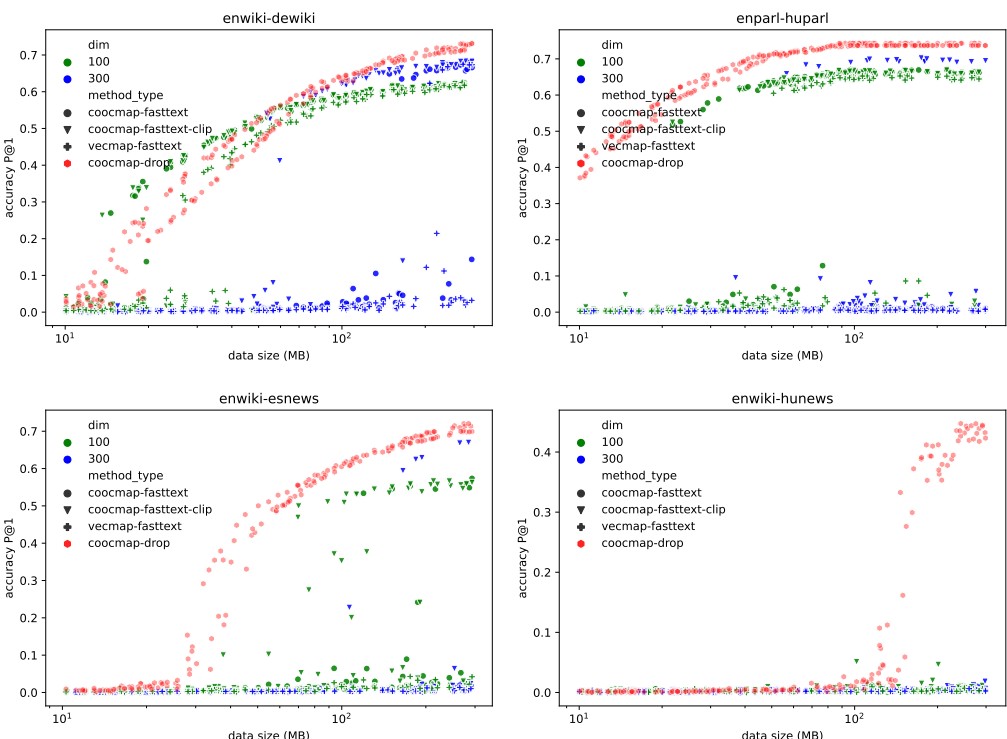

Figure 6: Improving fasttext with hyperparameter and clipping. Drop was set to $\min(20, 20\frac{d}{400})$ resulting in a double curve since $d = 100, 300$ were both used.

In the main results, we used default parameters, where the important ones were skigram, lr: 0.05, dim: 300, epoch: 5. In analysis section, we improve fasttext further by tuning dimension, learning rate and epoch number, then by using coocmap, clip and drop. The effect of dimension has been specifically explored in Figure 4. Here we show how 100 dimension and 300 dimensions as we vary with the size of data – 100-dim tend to be more data efficient whereas 300-dim tend to reach higher performance. The learning rate was slowed as $0.1(d/50)^{-1/2}$ to account for observed instability in higher dimensions. The epoch was increased to $5 \times (300/|D|)^{1/2}$ for data size $D$ in MB to run more epoch on smaller data size. We did not observe too much difference between this and default parameters beyond stablizing higher dimensions.

vecmap-fasttext only fully works for the two cases with no domain mismatch – enwiki-dewiki and enparl-huparl. In contrast, coocmap-fasttext show that fasttext contains the information to tackle harder cases of enwiki-esnews, showing that while fasttext is retaining the necessary information, it does not have the necessary linear algebraic property for vecmap to work. On enparl-huparl, vecmap-fasttext did not work for 300 dimensions whereas coocmap-fasttext-clip did using the same vectors with clipping, achieving better accuracy than vecmap-fasttext 100 dim. On enwiki-esnews, only coocmap-fasttext-clip and coocmap-fasttext worked. Finally, all fasttext based methods failed on enwiki-hunews.

It is always possible that we are still using fasttext poorly, but we covered the main factors and clarified the role of dimension which seems sufficient to explain the observed behaviors.

# C Clip, drop and truncate

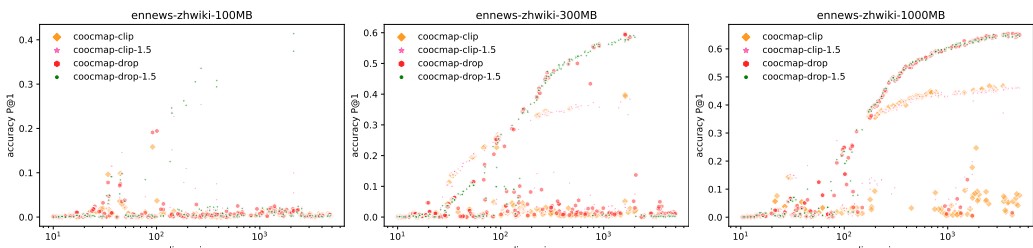

Figure 7: Accuracy vs. **dimension** on ennews to zhwiki, on 100MB, 300MB and 1000MB of data.

In this section, we try to push our most difficult case of ennews-zhwiki further by varying hyper-parameters as well as dimensions. In our main results in Figure 3, ennews-zhwiki barely had any transition before reaching high accuracy at 800MB, making us suspect there might be signal before that but the hyperparameter was not optimal. So here we try clip at 1.5% for **coopmap-clip-1.5, drop-1.5** instead of 1% as in every other experiment. We also adjust drop to $\min(20, 20\frac{d}{400})$ instead of a constant 20 so that less is dropped for lower dimensions $d$.

In Figure 7, we get the typical behavior of increasing accuracy with increasing dimension for 1000MB of data all the way to the end. For 100MB and 300MB, this only worked when we also truncate some of the tail end of the SVD. Even here, the maximum accuracy was reached at around 2000 dimensions out of 5000, reaching > 50% accuracy at 300MB of data and 2000 dimensions. It is also remarkable that while clipping more resulted in more stability, the final accuracies are virtually identical when both clip and clip-1.5 works or when both drop and drop-1.5 works. This is a case where it is helpful to truncate some of the tail of the SVD as well, at least for 100MB and 300MB where all failed at full dimensions. However, the accuracy still increased up to 2000 dimensions, much higher than typical vectors for the vocabulary size according to conventional wisdom.

# D More details and ablations

## D.1 Effect of initialization

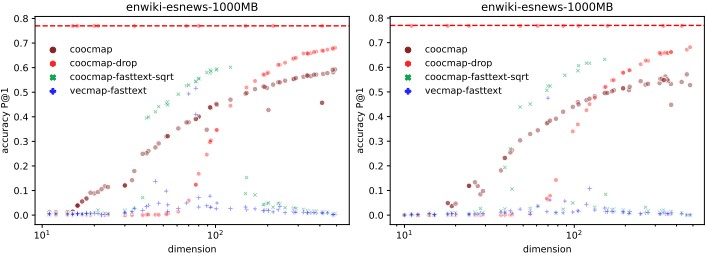

Figure 8: Accuracy vs. dimensions on enwiki-esnews. 1000MB of data. Left: use full coocmap initialization. Top red line is when coocmap-drop uses all 5000 dimensions. Right: use their own initialization.

To rule out that fasttext is only limited by the unsupervised initialization step, Figure 8 shows when all methods use the unsupervised initialization of full rank coocmap (left), vs their own unsupervised initialization (right). Here vecmap-fasttext and coocmap-fasttext-sqrt both worked for a slightly wider range of dimensions but the differences are small. Using the clipped version of initialization also did not make much difference. Overall, varying initializations did not matter too much for unsupervised initializations. On the other hand, for supervised initialization, coocmap tend not be very sensitive once there is enough data. vecmap can be more sticky and is more affected by supervised initialization, possibly because $W$ can reach a local minimum.

## D.2 Effect of matching method

The matching method was critical in getting coocmap to work, where simply applying the bidirectional matching does not work even on the simplest enwiki-eswiki. It starts to work in the first few iterations,

but then half of the vocabulary gets matched to the one word (often but not always [UNK]) and accuracy go to 0. Some hubness mitigation (Dinu et al., 2015) is a must for coocmap. This is not surprising since the indices $s, t$ can make arbitrary matches, whereas the vector space cannot make arbitrary matches from a rigid rotation.

We tried a naive and greedy mutual matching method that also solved the problem, but CSLS is better, simpler and is more similar to prior works. A proper matching methods such as Hungarian algorithm or softer matching method already tested on bilingual dictionary induction (Marchisio et al., 2022a) should work as well.

**Subwords.**   we used fasttext since it is more standard for this problem, and we also checked that the subwords information made no difference by turning them off in fasttext hyperparameters. This is reassuring for the findings to apply to word2vec as well.

## E   From vectors to association matrices

We derive a more precise relationship between coocmap and vecmap using the notations of Section 3.

Let the $d$-dimensional vectors be $X, Z \in \mathbb{R}^{|V| \times d}$ such that they are whitened $X^\mathsf{T} X = Z^\mathsf{T} Z = I_d$. Then for coocmap, the association matrices is $K(Y) = (YY^\mathsf{T})^{1/2} = YY^\mathsf{T}$. We can show that for dot product cdist, and permutations $s, t$,

$$\mathrm{cdist}(K(X)[:, s], K(Z)[:, t]) = \mathrm{cdist}(XW, Z) \tag{5}$$

where the LHS is the distance function in coocmap of Alg 1 and the RHS is the distance function in vecmap of Alg 2. To see this,

$$
\begin{aligned}
&\mathrm{cdist}(K(X)[:, s], K(Z)[:, t]) \\
&= K(X)[:, s] \cdot K(Z)[:, t]^\mathsf{T} \\
&= (XX[s])^\mathsf{T} \cdot (ZZ[t]^\mathsf{T})^\mathsf{T} \\
&= XX[s]^\mathsf{T} \cdot Z[t]Z^\mathsf{T} \\
&= X(X[s]^\mathsf{T} Z[t])Z^\mathsf{T} && \text{if } X[s]W = Z[t], \text{ then } W = (X[s]^\mathsf{T} X[s])^{-1} X[s]^\mathsf{T} Z[t] \\
&= XWZ^\mathsf{T} && \text{since } (X[s]^\mathsf{T} X[s])^{-1} = I, \ W = X[s]^\mathsf{T} Z[t] \\
&= \mathrm{cdist}(XW, Z)
\end{aligned}
$$

We can try relaxing the whitening assumption. w.l.o.g., let $X = U_1 \Sigma_1$, $Z = U_2 \Sigma_2$ where $U_i^\mathsf{T} U_i = I_d$. This can be obtained via SVD on any $X' = U_1 \Sigma_1 V_1^\mathsf{T}$ so that the association matrix is not affected, i.e. $K(X') = (X'X'^\mathsf{T})^{1/2} = (XX^\mathsf{T})^{1/2} = U_1 \Sigma_1 U_1^\mathsf{T}$. In this case, $W = U_1[s]^\mathsf{T} U_2[t]$ is the mapping that would make the two sides equal.

However, if we try least square solve as before, we get $W = \Sigma_1^{-1} U_1[s]^T U_2[t] \Sigma_2$ . The Procrustes solution for $X[s]W = Z[t]$ (or $U_1[s]\Sigma_1 W = U_2[t]\Sigma_2$) is $W = \mathrm{svd\text{-}norm}(\Sigma_1^\mathsf{T} U_1[s]^\mathsf{T} U_2[t] \Sigma_2)$. svd-norm takes the SVD but set all singular values to 1. Neither will make the two sides equal exactly but these have some similarities.

## F   What is in higher dimensions?

Although the increasing accuracy, the absolute results and the consistency when we keep higher dimensions suggest that there are useful signals, it could always be that we are using lower dimensions poorly or benefting from other unknown effects. In this section, we compare the full-rank matrix and the low-rank matrix to see what exactly is the difference. If the information lost by low rank seems useful or robust, then we can be more confident in retaining this information. Indeed, the low rank version seem to lose or de-emphasize a lot of world knowledge, specific phrases and the like while making the association matrix more like a similarity matrix. In this section, we compare the full-rank matrix and the low-rank matrix to see what exactly is the difference. If the information lost by low rank looks useful or robust, then we can be more confident in retaining this information. Indeed, the low rank version seem to lose or de-emphasize a lot of world knowledge, specific phrases and the like while making the association matrix more like a similarity matrix.

We construct the full dimensional matrix, apply normalize and drop. Then we use the thresholds established by clip. If an entry needs to be clipped in the full-rank matrix, but somehow is already

reduced and no longer need to be clipped in the lower-rank matrix, we record this as **full-rank+** in Table 9. Conversely, if low-rank instead increased an entry so that it should be clipped, we note it as **300-dim+**.

From these examples, the differences are clear. The full dimensional matrix contain more world knowledge and otherwise favor dissimilar words that are highly related: *richard-nixon (famous person), portland-oregon maine (geography), error-margin, molecular-weight (scientifc term), tokyo-1964 (olympic), basketball-jordan james (famous players), cbs-60 minutes (show on CBS), louisiana-parish purchase (famous event).*

In contrast, the 300-dim+ favors identical or similar words like *state-state, age-age, who-whom, truth-true, tokyo-pyeongchang, cbs-fox, family-households*, making the association matrix more like the similarity matrix. This is also intuitive from SVD as well: if the association matrix is $USV^\intercal$, then the similarity matrix is $US^2V^\intercal$ whereas low dimension is $US_dV^\intercal$ with both $S^2$ and $S_d = \mathrm{trunc}_d(S)$ emphasizing higher singular values.

We also tested full-rank vs 1000-dim, where the effects are similar to the full-rank vs 300-dim shown, but more subtle and harder to present clearly. When we tested drop we found drop tend to reduce associations between pairs of numbers, words and punctuations and the like, whereas it increased associations between article and words. It is not clear if this is good or bad.

## G   Example predictions

We include some predictions for extra information. The P@1 is quite generous to small vocabulary $V$, since it does not evaluate on entries if the source is not in $V_1$, or if no targets are in the $V_2$, treating the evaluation as out-of-vocabulary instead. So we show the number of correct predictions and the source overlap with the reference dictionary, both out of 5000. Results are shown in Table 10.

| source | full-rank+ | source | full-rank+ |
|---|---|---|---|
| **NewsCrawl** | | **Wikipedia** | |
| richard | nixon | families | residing |
| cooper | anderson | called | so |
| anderson | cooper | aspect | ratio |
| report | contributed | error | margin |
| at | least | walter | scott |
| no | longer | yet | another |
| than | rather | oregon | portland |
| this | earlier | scott | walter |
| dean | wells | unless | otherwise |
| kyle | walker | usa | today |
| crazy | rich | jones | indiana |
| behind | scenes | louisiana | parish |
| walker | kyle | british | columbia |
| politically | correct | water | %, |
| thomas | cook | cooper | alice |
| graham | billy | maryland | baltimore |
| sense | common | why | reason |
| detroit | lions | illinois | chicago |
| correct | politically | lord | rings |
| aaron | rodgers | james | bond |
| 2000 | russell | siege | laid |
| star | wars | %). | persons |
| light | shed | molecular | weight |
| town | hall | then | since |
| almost | certainly | bob | hope |
| hollywood | reporter | if | even |
| texas | austin | coast | guard |
| cash | flow | final | fantasy |

| full-rank+ | 300-dim+ |
|---|---|
| **source: tokyo** | |
| **NewsCrawl** | |
| japanese | winter |
| prosecutors | south |
| electric | china |
| metropolitan | korea |
| detention | pyeongchang |
| **Wikipedia** | |
| bay | chinese |
| 1964 | china |
| 2020 | korea |
| ward | areas |
| stock | san |
| rose | taiwan |

| full-rank+ | 300-dim+ |
|---|---|
| **source: cbs** | |
| **NewsCrawl** | |
| face | fox |
| 60 | night |
| nation | channel |
| boston | telling |
| minutes | abc |
| **Wikipedia** | |
| 60 | bbc |
| minutes | disney |
| columbia | cable |
| walter | live |
| rather | shows |

| full-rank+ | 300-dim+ |
|---|---|
| **source: truth** | |
| **NewsCrawl** | |
| speak | true_ |
| commission | are |
| moment | than |
| post | be |
| uncomfortable | story |
| **Wikipedia** | |
| tell | god |
| commission | view |
| table | than |
| functional | universe |
| search | which |

| full-rank+ | 300-dim+ |
|---|---|
| **source: basketball** | |
| **NewsCrawl** | |
| operations | club |
| corruption | rugby |
| division | nfl |
| courts | clubs |
| magic | golf |
| james | ball |
| **Wikipedia** | |
| nba | club |
| operations | stadium |
| country | ice |
| memorial | rugby |
| college | clubs |

Figure 9: Left: word pairs favored by the full rank matrix (full-rank+) on NewsCrawl or Wikipedia sorted by the amount of difference over all pairs of words. 300-dim+ not shown since 72/top 100 consists of identical words with the rest mostly synonyms. Right: same comparison but for selected words, also sorted by the difference.

| | ennews | zhwiki | | enwiki | esnews | | enwiki | dewiki |
|---|---|---|---|---|---|---|---|---|
| 4458 | swift | 冫 | 3484 | victims | víctimas | 2732 | looking | ergaben |
| 947 | significant | 重大 | 2727 | somewhat | bastante | 3056 | deputy | stellvertreter |
| 2179 | strength | 力量 | 682 | star | estrella | 1805 | 1977 | 1977 |
| 2972 | letters | 信 | 2847 | centers | centros | 337 | september | september |
| 2401 | coverage | 直播 | 4176 | theology | sindicatos | 4669 | strict | streng |
| 733 | idea | 提议 | 868 | historic | históricos | 2248 | sector | bereich |
| 4260 | assist | 协助 | 451 | research | investigaciones | 414 | came | kam |
| 2803 | properties | 特性 | 3548 | tell | comprobar | 105 | while | hingegen |
| 4542 | circuit | 法官 | 2375 | script | estuvieron | 86 | county | county |
| 4191 | saint | 圣 | 3570 | elevation | mínima | 702 | pacific | bewohnern |
| 612 | recently | 近年 | 3021 | 800 | 600 | 248 | house | haus |
| 3360 | maria | 娜 | 765 | forms | formas | 1178 | product | produkt |
| 1846 | surprise | 在内 | 4575 | delayed | adelantado | 164 | second | zweite |
| 713 | copyright | & | 731 | r | r | 1907 | contained | enthielt |
| 2015 | carrying | 携带 | 4058 | accurate | precisa | 3208 | easy | schwer |
| 3843 | suspects | 自杀 | 3708 | technologies | tecnologías | 3200 | tone | ton |
| 1536 | dropped | 下降 | 854 | elements | elementos | 4578 | consensus | zustande |
| 1014 | largest | 最大 | 3187 | abbey | estás | 477 | television | fernsehen |
| 2672 | communica | 通讯 | 1483 | mountains | sierra | 1716 | 75 | 75 |
| 4934 | masters | 公开赛 | 4708 | southwestern | básico | 4489 | handed | übergeben |
| 989 | words | 词 | 1094 | problems | problemas | 2866 | electrical | elektrische |
| 4400 | pan | 亚 | 3813 | compete | competir | 2976 | proposal | vorschlag |
| 3289 | maximum | 以上 | 934 | remains | sido | 4788 | beta | )- |
| 1620 | ceo | 总裁 | 1642 | crown | corona | 691 | food | nahrung |
| 821 | offer | 提供 | 3045 | harbor | sentirse | 251 | power | macht |
| 2611 | 150 | 150 | 4274 | phone | teléfono | 725 | able | konnte |
| 1520 | individual | 个人 | 3626 | z | subraya | 286 | late | späten |
| 801 | cancer | 疾病 | 302 | german | alemán | 290 | park | park |
| 4003 | ."" | 任内 | 133 | early | naranja | 3903 | en | en |
| 319 | always | 总是 | 4955 | provisions | vigente | 808 | source | quelle |
| 4424 | backs | 司职 | 3366 | fruit | aceite | 1050 | cost | kosten |
| 1345 | modern | 现代 | 2249 | engines | motores | 1803 | 99 | 99 |
| 4835 | spell | 词 | 4508 | trump | trump | 2179 | stand | stehen |
| 4737 | loud | 声 | 2669 | forming | formar | 3926 | pitch | ton |
| 3894 | metres | 公尺 | 764 | personal | personales | 4449 | populated | besiedelt |
| 4472 | flood | 洪水 | 1810 | identity | identidad | 3979 | hunt | jagd |
| 4648 | attract | 吸引 | 782 | strong | fuerte | 4909 | wore | trugen |
| 3297 | lowest | 最低 | 936 | recent | recientes | 2510 | seems | scheint |
| 539 | 50 | 50 | 955 | always | siempre | 3978 | cousin | bruder |
| 1083 | build | 建造 | 72 | into | nos | 2325 | operate | betreiben |
| | correct | 1411 | | correct | 2024 | | correct | 2424 |
| | source | 3341 | | source | 4351 | | source | 4401 |
| | P@1 | 66.30% | | P@1 | 71.60% | | P@1 | 77.05% |

Figure 10: 40 random samples from all predictions. Source indices are included to reflect frequency where smaller means more frequent in the source language (left columns).

