# OpenReview forum: "Accessing Higher Dimensions for Unsupervised Word Translation"
_NeurIPS.cc/2023/Conference — NeurIPS 2023 poster_

### Official Review · Reviewer_LMyn · 2023-06-26

**Soundness:** 4 excellent
**Presentation:** 4 excellent
**Contribution:** 3 good
**Rating:** 7
**Confidence:** 4

**Summary:**

The authors argue that relying solely on low-dimensional vectors misses out on better denoising methods and valuable world knowledge present in high dimensions. Therefore, they propose coocmap, a method that can exploit high-dimensional co-occurrence counts. Through extensive experiments, the authors show coocmap works very well under different data sizes and data from various domains.

**Strengths:**

- The paper is well-presented and easy to follow.
- The figures are illustrative and informative.
- The authors conduct extensive experiments on different language pairs, including similar and dissimilar languages. Besides, the authors also explore the influence of data sizes and domain mismatch.
- The method proposed, coocmap, is straightforward but effective. Compared with some baselines, the method seems to work surprisingly well on small data sizes.


**Weaknesses:**

There is no apparent weakness in this paper. However, there are some questions/suggestions (see below).

**Questions:**

- Line 12: "similar data", the author should specify similar in terms of what?

- As vocabulary size is an important hyperparameter of the method. The authors should mention it in the main content (probably in the First paragraph in Section 4). The authors are also encouraged to explore the influence of vocabulary size on performance.

- The authors should provide more intuition on why clip and drop are very helpful for harder cases, e.g., en-zh. As the vanilla coocmap is not stable on en-zh.

- Line 194: "NewsCrawl 2019.es and ..." -> remove "and"

- Figure 2: It would be great if the authors could say the upper 3 figures are similar languages while the lower three are less similar languages in the caption.

- I have a more general question: the study itself is interesting and demonstrates how a simple higher-dimensional co-occurrence matrix could largely help unsupervised word translation. However, in the era of LLMs, when and where can the method be possibly applied?

**Limitations:**

The authors discussed the limitation and broader impact of the paper. It is encouraged to include some additional experiments to explore the influence of vocabulary size.

---

> ### Author Rebuttal · Authors · 2023-08-09
>
> Thank you for the helpful review and corrections, which will help us improve the revision.
>
> > Line 12: "similar data", the author should specify similar in terms of what?
>
> We will specify similar in domains.
>
> > The authors should provide more intuition on why clip and drop are very helpful for harder cases, e.g., en-zh. As the vanilla coocmap is not stable on en-zh.
>
> Thanks for the suggestion, we will discuss more. Clip can be motivated by robust statistics and some clipping happens automatically for SVD truncation. We also motivated clip intuitively in the section it was introduced but not specifically for harder cases. As for drop, we provided two citations that did it to a lesser extend and we admittedly do not understand drop as well as clip. One additional intuition is that drop already happened for free in word vectors.
>
> > As vocabulary size is an important hyperparameter of the method. The authors should mention it in the main content (probably in the First paragraph in Section 4). The authors are also encouraged to explore the influence of vocabulary size on performance.
>
> We will add a summary of the main points about vocabulary size to the first paragraph of Section 4 in addition to reference to limitations.
>
> > the study itself is interesting and demonstrates how a simple higher-dimensional co-occurrence matrix could largely help unsupervised word translation. However, in the era of LLMs, when and where can the method be possibly applied?
>
> It is easier to use the findings than the method directly. Retrieval augmented LMs and knn LMs are perhaps two places where the findings may apply. Transformer LMs still have a low dimensional embedding layer which is typically a few thousand dimensions for each subword. Previous work have shown that knn and retrieval can improve the perplexity, but did not explain why. Our results suggest that the information lost in dimension reduction is probably why retrieval can be beneficial. Note that retrieval is just indexing for higher-order co-occurrences and raise the possibility that simply having the full dimensional co-occurence may also be helpful and capture information not already captured by transformer models.
>
> It would be pretty interesting to see if the drop and clip techniques are applicable in LLMs directly since we show that even low-dimensional vectors can benefit from clip, which suggests that other activations that approximates co-occurences may also benefit from clipping. On the other hand, drop was needed to actually get the improvement with higher dimensions, which may also be helpful to projects attempting to augment LLMs with a sparse high dimensional object.

---

> > ### Comment · Reviewer_LMyn · 2023-08-18
> >
> > Thank you very much for your detailed response. All my questions have been answered now.

---

### Official Review · Reviewer_AbFD · 2023-07-04

**Soundness:** 3 good
**Presentation:** 2 fair
**Contribution:** 4 excellent
**Rating:** 7
**Confidence:** 2

**Summary:**

This paper proposes using high-dimensional co-occurrence statistics for unsupervised word translation rather than relying on low-dimensional vector embeddings. They propose using coocmap (uses association matrix of cooccurrences) and combining this with regularization operations (clip/drop) that eliminate noise. The goal is to achieve good performance on this task more efficiently (less data+compute) and good cross-domain performance.

Experiments are performed on 7 languages and 3 domains, measuring precision@1 on 5K most frequent words against the MUSE dictionary. The baseline uses fasttext vectors. Coocmap+variants outperform fasttext vectors across languages that are not very similar and across domains in some cases using an order of magnitude less data. Analysis shows that high-dimensional co-occurrence data is more robust than its low-dimensional counterparts.


**Strengths:**

1. This work is interesting and identifies that for the task of unsupervised word translation the raw learning signal generalizes better than compressing into low-dimensional vectors which are lossy.
2. The domain mismatch experiments perhaps highlight this point well by relying on more fine-grained information that may be lost in the embeddings.
3. The proposed denoising functions are clever and simple.


**Weaknesses:**

The current draft is at times not very articulate and it can take a few passes to understand the intended point. Some revisions might help!

Is the following an accurate characterization of the work: low-dimensional representations are not expressive enough for this task so we start by overfitting with cooccurrences and use some heuristics to regularize?
If yes, then does this feel more like gaming the task? And would that make the claim about being able to transfer to other monolingual tasks unfair? Doesn’t seem too convincing yet.

Scaling does still seem like a challenge: would switching to BPE preserve the multilingual distributional hypothesis?

Nit: including low-resource languages in the experiments might make the paper stronger.


**Questions:**

Could some of the headroom in the vectors be covered by distilling from coocmap to low-dimensional vectors? i.e. is it an optimization issue or a capacity issue?

**Limitations:**

See above for concerns regarding transferring to other tasks and being able to scale easily.

---

> ### Author Rebuttal · Authors · 2023-08-09
>
> Thanks for the review and feedback. We are glad that you liked the content of the paper and we will try harder to make the paper more articulate in future revisions.
>
> > Is the following an accurate characterization of the work: low-dimensional representations are not expressive enough for this task so we start by overfitting with cooccurrences and use some heuristics to regularize?
>
> > If yes, then does this feel more like gaming the task? And would that make the claim about being able to transfer to other monolingual tasks unfair? Doesn’t seem too convincing yet.
>
> This is not accurate. As there is no training data to fit but only unlabeled corpus, there is no overfitting here and no regularization. As for expressivity, low-dimensions representations is how previous work solved unsupervised word translation, showing the power and ease-of-use of low-dimensional vectors. However the community then draws the wrong conclusion from this success, concluding that vectors are also better in every way and there is only noise in high-dimensions to be removed. We show that a few simple processing steps actually allow us to use the full high-dimensional co-occurrences, leading to similar and then better results than low-dimensional vectors.
>
> No, we could not game the task and do not see a good way to game it as the task is fully unsupervised.
> Speculations were clearly labelled and our reasoning is this:
> 1) naively using high dimension may indeed lead to conclusions like in [1] because no denoising was used at all
> 2) here is a way to use the higher dimensional information for unsupervised MT so that higher dimension is helpful
> Thus we speculate that the actual situation is that the necessary but simple denoising operations were not used in previous work, which leads to the conclusions in [1]. This proposed explanation is actually much more intuitive than the current "vectors make all NLP tasks better". The reviewer can draw their own conclusions about this, but the author is willing to bet on it. We will try to make this more clear.
>
> [1]: Baroni et al. Don’t count, predict! A systematic comparison of context-counting vs. context-predicting semantic vectors
>
> > Scaling does still seem like a challenge: would switching to BPE preserve the multilingual distributional hypothesis?
>
> Scaling coocmap to more data is easier than with word vectors. Scaling to more word types is more challenging than using word vectors, but some incremental approach (limiting the search space to words of similar frequency) and approximations (including truncation and vectorization) can be used.
>
> BPE with large enough vocabulary should preserve at least some of the multilingual distributional hypothesis but probably not as well as words, which are betters unit for translation than subwords. BPE can be combined into coocmap by featurization.
>
> > Could some of the headroom in the vectors be covered by distilling from coocmap to low-dimensional vectors? i.e. is it an optimization issue or a capacity issue?
>
> Experiments included in the paper show the gap is probably more of a "capacity" issue, in the sense that low-dimensional vectors lost useful information. In the experiment of coocmap-fasttext vs coocmap in figure 4, the same optimization method was used, only their "capacity" differs.
>
> Not sure how distilling applies here exactly, but word vectors are probably doing as good a job as they can given their low dimensions. We showed their superiority over SVDs at the same dimension.

---

> > ### Comment · Reviewer_AbFD · 2023-08-17
> > **Author response**
> >
> > I should have used quotes---"overfitting" and "regularization"---to highlight that I meant this less as precise mechanics and more as a loose analogy...rather than lossy compression as in low-dimensional vectors, this approach starts out with all coocurrence information and uses clip/drop etc. to remove the noise. Anyway, I buy that this is a clever approach for this task but am not yet convinced about robustly transferring to and generalizing for more complex tasks---perhaps you're right and the lost fine-grained information is helpful for generalization..
> > Either way, the paper was an interesting read and hope the revisions can make it a bit easier to follow.
> >
> > Thanks for your response. I still recommend accepting.

---

> > > ### Author Response · Authors · 2023-08-17
> > >
> > > Thanks for the continued discussions and great to hear you found the paper interesting.
> > >
> > > > perhaps you're right and the lost fine-grained information is helpful for generalization.
> > >
> > > Both vectors and drop/clip lose some information. However d-dimensional vectors lose a lot more information by parameter count than drop and clip. In summary:
> > >
> > > * full: V^2
> > > * d-dim vectors: V d
> > > * clip: V^2, but epsilon of largest are now identical
> > > * drop k: (V-k) * V
> > >
> > > > not yet convinced about robustly transferring to and generalizing for more complex tasks
> > >
> > > Personally I'm willing to bet that this will generalize to other word vector evaluations that may be past their prime as well. For the important task of language modeling, transformers already use much higher and increasing dimensions to represent subwords. They still might benefit from even higher dimensions, but it is less clear if the increased compute will be worth it. On this point it is natural not to be convinced based on this paper! Curious if there is any particular complex task that you would be interested to see.

---

### Official Review · Reviewer_RpPs · 2023-07-07

**Soundness:** 3 good
**Presentation:** 2 fair
**Contribution:** 3 good
**Rating:** 6
**Confidence:** 3

**Summary:**

This paper introduces coocmap, an approach for unsupervised word translation that uses high-dimensional co-occurrence statistics instead of lower-dimensional word embeddings. The approach is generally analogous to vecmap, alternating between distance computations and a matching phase. Coocmap is often more data efficient than other methods, and its performance may be improved by regularizing the co-occurence matrix (normalization, value clipping, dropping large singular vectors).

**Strengths:**

The proposed approach goes against the conventional wisdom by using high-dimensional co-occurrence vectors instead of word embeddings. The method is generally more data efficient than previous work, which may be surprising.

The paper describes strategies to improve upon using the raw co-occurrence matrices directly.

The experiments cover multiple languages (although English is part of every pair) and domains.

**Weaknesses:**

Given that unsupervised word translation has been introduced many years ago, but arguably has limited practical applications, the authors should more clearly motivate their work.

In addition to data efficiency, final accuracy performance should also be discussed (although we can read it from the figures).

It might have been interesting to analyze how well the approach works based on word frequency (and with vocabulary sizes beyond 5000).

The paper can be difficult to read at times. vecmap should arguably be described before coocmap, although presenting them in parallel may still be acceptable.

The paper slightly exceeds the 9-page limit.

**Questions:**

Why use 50% as a threshold for "works"?

Given that you use high-dimensional co-occurrence statistics, what are the compute and memory requirements compared to approaches that use low-dimensional vectors?

[L45] retain -> retaining

[L48] have -> having

[L286] Fragment "Actually showing the same behaviors as fasttext."

[L292] retain

[L303] modeling (or modelling, or "to model")

[L336] focuses

**Limitations:**

Yes, the authors discussed limitations of their work.

---

> ### Author Rebuttal · Authors · 2023-08-09
>
> Thank you very much for the review and concrete corrections. We will make the corrections in the final version.
>
> > Given that unsupervised word translation has been introduced many years ago, but arguably has limited practical applications, the authors should more clearly motivate their work.
>
> The particular MUSE evaluation was indeed introduced "many years ago" and the underlying task of word translation has been around for even more years, which shows how intuitive the task is. Our main contribution is to show that unsupervised translation is also possible without word vectors and that the data requirement is surprisingly little using a simple method.
>
> More generally, as LLMs clearly demonstrated unsupervised abilities, future MT systems are likely to use more unsupervised data. The path from research to practical application may take time and immediate practical application should not be a requirement for research.
>
> > In addition to data efficiency, final accuracy performance should also be discussed (although we can read it from the figures).
>
> The final accuracy was judged not to be unreliable as several previous work warned against taking this benchmark too seriously. On the other hand, MUSE is absolutely good enough to tell if the unsupervised method is successful at all.
>
>
> > It might have been interesting to analyze how well the approach works based on word frequency (and with vocabulary sizes beyond 5000).
>
> Yes, though we needed a cutoff to develop the method quickly and keep it simple. An incremental approach where the most common words is solved first then extending the search to larger vocabulary is fairly promising but still introduces complications in search strategy and implementation. This point is stated in limitations and we will emphasize more.
>
> > The paper slightly exceeds the 9-page limit.
>
> Thanks for pointing this out and for overlooking this mistake, we followed instructions that "additional pages containing only broader impact statement and references are allowed", which seems outdated. Will fix for the final version.
>
> > Why use 50% as a threshold for "works"?
>
> This is just an arbitrary threshold so we can measure the data requirement. Since the accuracy is never 100%, and are binary in that they are either near 0% or is eventually greater than 60%. It would have been fine to pick anywhere between 10% and 60% to establish the comparison.  One analogy supporting the choice of 50% is the half-life of an exponential decay, which is admittedly more precise than our use case.
>
> > Given that you use high-dimensional co-occurrence statistics, what are the compute and memory requirements compared to approaches that use low-dimensional vectors?
>
> This is discussed in limitations for compute as a function of vocabulary size V, which favors d-dimensional vectors by a factor of V/d. For memory, both methods are O(V^2). In term of data size scaling, both methods are constant.

---

> > ### Comment · Reviewer_RpPs · 2023-08-21
> >
> > Thank you for your response. I am inclined to increase my rating from 5 to 6.
> >
> > > The path from research to practical application may take time and immediate practical application should not be a requirement for research.
> >
> > I agree that this is not strictly necessary, but it could increase the impact of the paper.
> >
> > >The final accuracy was judged not to be unreliable as several previous work warned against taking this benchmark too seriously
> >
> > In the camera-ready version, you could mention this and cite the papers you refer to here (some of which may already be in the references).
> >
> > > Will fix for the final version.
> >
> > If the paper is accepted, you should be allowed a 10th content page.

---

### Official Review · Reviewer_ZKax · 2023-07-07

**Soundness:** 3 good
**Presentation:** 3 good
**Contribution:** 3 good
**Rating:** 5
**Confidence:** 3

**Summary:**

This paper proposes coocmap, a method for unsupervised word translation that uses high-dimensional co-occurrence statistics instead of low-dimensional vectors. The authors show that relying on low-dimensional vectors can lead to suboptimal denoising methods and overlook useful world knowledge in high dimensions. The authors demonstrate that unsupervised translation can be accomplished with a smaller amount of data and in a wider range of scenarios than previously believed. They also suggest that co-occurrence-based methods may outperform low-dimensional vectors in other tasks.

**Strengths:**

1.	This paper successfully demonstrates the effectiveness of high-dimensional co-occurrence statistics and provides empirical evidence that unsupervised translation using only co-occurrence statistics is feasible.
2.	The authors provide a detailed comparison with vecmap and present coocmap in a clear and easy-to-follow manner.
3.	The paper showcases the trend of retrieval accuracy during the training process on the data size, which is valuable for understanding the training dynamics of word translation capacity.


**Weaknesses:**

1.	The experiment setup in the paper is somewhat ambiguous, as the authors state that the experiment was conducted on the top 5000 most common words in each language using the MUSE dictionary, but do not provide a clear definition of what they mean by "most common." Additionally, the MUSE dictionary contains a significant number of "identical translation" pairs, where the source word and corresponding target word are the same. It is unclear how these types of translation pairs were processed in the experiment.
2.	The method is only compared with vecmap. Maybe more baseline methods such as LNMAP[1], FIPP[2] and [3] should be included for a comprehensive understanding of model performance.
3.	The paper could benefit from a more comprehensive discussion of prior works in the field, which would help readers contextualize the authors' contributions. It is unclear whether this is the first paper to use co-occurrence statistics to induce an unsupervised word translation dictionary, and it would be helpful to know if there are any previous works in this area.

[1]LNMap: Departures from isomorphic assumption in bilingual lexicon induction through nonlinear mapping in latent space.
[2]FILTERED INNER PRODUCT PROJECTION FOR CROSSLINGUAL EMBEDDING ALIGNMENT
[3] Improving Word Translation via Two-Stage Contrastive Learning


**Questions:**

1.	In Figure2, there is a sharp surge of performance for vecmap-fasttext on enwiki-frwiki and enwiki-eswiki. Can you explain this?

**Limitations:**

see weakness

---

> ### Author Rebuttal · Authors · 2023-08-09
>
> Thanks for the review, we are glad to hear that you found the paper easy to follow. In particular, we claim that we are the first to do unsupervised word translation with just co-occurrences and there are several difficulties with doing this which we identified in the paper. More details in the answer to the particular question, please do reconsider your score as this is an important claim.
>
> > The experiment setup in the paper is somewhat ambiguous, as the authors state that the experiment was conducted on the top 5000 most common words in each language using the MUSE dictionary, but do not provide a clear definition of what they mean by "most common."
>
> The most common words is based on word frequency in the corpus and returned by the WordLevel tokenizer. In python, `Counter(corpus).most_common(5000)` gets the 5000 most common words when `corpus` is a list of words. For example, the most common word in `corpus=['cat', 'cat', 'dog']` would be `{'cat'}`.  We will add "by word frequency" to further clarify.
>
> > Additionally, the MUSE dictionary contains a significant number of "identical translation" pairs
>
> Yes indeed identical translation is a known issue with the evaluation set. We avoided the worst cases for Chinese / English by filtering out all Latin characters from the Chinese corpus, so there is at least no identical words for the en-zh pair. For other languages where words may actually be identical, we ignore this issue but track how many identical pairs were matched in the experiments. This and other problems with the benchmark is only important if we claim that small improvements on MUSE are an important contribution, which we do not. On the other hand, the MUSE evaluation is good enough to establish if unsupervised translation is happening at all and at how much data is needed, which is our main evaluation.
>
> > The method is only compared with vecmap... more baseline methods such as LNMAP[1], FIPP[2] and [3] should be included
>
> Because establishing acc(coocmap) > acc(other method) is not a main point of the paper. All of the cited methods are also vector space methods that depends more on which vectors were used and on what data they are trained on. Those methods also compared with vecmap in their own papers. We share your concerns that MUSE/other word translation evaluations should not be taken too seriously, and we do not use MUSE to claim that method X has slightly higher accuracy than method Y. Instead we use MUSE to test which method works at all and how much data was needed. Note that we do not even claim that acc(coocmap) > acc(vecmap), as vecmap with enough data outperformed the basic coocmap as shown in Figure 2. It is only with "drop" and high dimensions that coocmap has higher accuracy than vecmap where drop changes the inputs but not the method.
>
> We compared to other normalization methods and other word vectors in the appendix, which was interesting but did not make the main paper.
>
> > It is unclear whether this is the first paper to use co-occurrence statistics to induce an unsupervised word translation dictionary, and it would be helpful to know if there are any previous works in this area.
>
> Thanks for the feedback. As far as we know, this is indeed the first paper to use occurrences to do unsupervised word translation successfully. We cited works on previous attempts that ultimately did not work unsupervised on general data. These previous works are cited prominently in the first sentence of introduction (Rapp 1995, Ravi and Knight 2011) and with more details in the discussions section and appendix. Rapp 1995 suggests that this might be possible but did not have the search methods. Ravi and Knight actually used parallel data in narrow domain like time expression and subtitles for their method to work. Their method also does not include the normalization and relative measurements that were crucial for coocmap to work.
>
> In fact, we probably could not do this successfully without knowing that it is possible and without some of the advances from vector based methods such as normalization and relative measurements, but vectors themselves were not necessary.
>
> > In Figure2, there is a sharp surge of performance for vecmap-fasttext on enwiki-frwiki and enwiki-eswiki. Can you explain this?
>
> All methods need a sufficient amount of data to work and the transition is usually fairly sharp. The intuition is once you can figure out an initial amount of word translations, then other frequent enough words should be easy to translate based on context.

---

> ### Comment · Reviewer_ZKax · 2023-08-21
> **Thanks for the rebuttal**
>
> I read the rebuttal and concerns from other reviewers as well. I keep my recommendation score.

---

> > ### Author Response · Authors · 2023-08-21
> >
> > Did we address the weaknesses raised in the review? We believe weakness 1 and weakness 3 were completely addressed and an argument was made about weakness 2. In particular it seems addressing weakness 3 could be important enough to improve the recommendation.

---

### Official Review · Reviewer_VTy1 · 2023-07-08

**Soundness:** 3 good
**Presentation:** 2 fair
**Contribution:** 3 good
**Rating:** 6
**Confidence:** 2

**Summary:**

In this paper, the authors target to solve unsupervised word translation, also called lexicon or dictionary induction, using their proposed method, coocmap, a simplified version of the conventional way, vecmap. Different from vecmap, coocmap can estimate word mapping without using rotation weights for row vectors. Instead, coocmap uses an association matrix to represent source and target word relations. Thus, coocmap can estimate source and target word mappings by rearranging the columns of the association matrix to reflect the updated word mappings during training time. Experimental results on language pairs from English to Spanish, French, German, Hungarian, Finnish, and Chinese show that coocmap outperforms vecmap, especially when the data size or dimension size increase.

**Strengths:**

- The proposed method, coocmap, outperformed the conventional approach, vecmap, in word translation accuracy on language pairs from English to Spanish, French, German, Hungarian, Finnish, and Chinese.
- Coocmap does not require a rotation matrix for projecting word embeddings to the ones of other languages; thus, it's simple.

**Weaknesses:**

- Not using the rotation matrix means coocmap cannot handle unseen words because their representation is not included in the association matrix of coocmap. Thus, the versatility of coocmap is less wide compared with vecmap.
- Many translation tasks require translations for texts rather than word translation. Thus, investigation of downstream tasks, especially for machine translation, is necessary to claim the advantage of coocmap even though it's not conducted in the paper.
- The motivation for using the unsupervised method needs to be clarified. In practice, we can use the MUSE [1] dictionaries you used to evaluate your approach, to train word translation models in a supervised manner.

[1] Lample, G., Conneau, A., Ranzato, M. A., Denoyer, L., & Jégou, H. (2018, February). Word translation without parallel data. In International Conference on Learning Representations. (paper: https://openreview.net/forum?id=H196sainb, code:https://github.com/facebookresearch/MUSE)

**Questions:**

- In the evaluation, why didn't you compare the performance of your coocmap with the reported scores of conventional methods? You can use Semeval 2017 [2] data for this purpose, similar to the paper of MUSE dictionaries [1] you used.
- Could you expand your approach to using subwords? This is because current natural language generation tasks, including machine translation, heavily rely on subwords rather than words. This part relates to the contribution of your work to downstream tasks.
- In the current draft, you use MB to show the size of the data. How did you calculate them? The data you used is compressed or not?
- It's difficult to estimate the computational complexity or space of both coocmap and vecmap by MB of the used data. Showing the vocabulary size is more helpful for readers to understand.

[2] Camacho-Collados, J., Pilehvar, M. T., Collier, N., & Navigli, R. (2017, August). Semeval-2017 task 2: Multilingual and cross-lingual semantic word similarity. In Proceedings of the 11th international workshop on semantic evaluation (SemEval-2017) (pp. 15-26). (https://aclanthology.org/S17-2002/)

**Limitations:**

You need to include the treatment of the unseen words in the limitation part of your paper. Furthermore, you need to refer to the fact that current translation models commonly use subwords, not words.

---

> ### Author Rebuttal · Authors · 2023-08-03
>
> Thanks for the review with many concrete points that may help us avoid possible confusions. We first address the weaknesses and then the questions. We emphasize that the most substantial proposed weakness about unseen word is wrong.
>
> > Not using the rotation matrix means coocmap cannot handle unseen words... versatility  less wide compared with vecmap
> > You need to include the treatment of the unseen words in the limitation part of your paper.
>
> There is no difference in the treatment of unseen words. For words unseen during matching process but seen in the corpus, you can use the occurrences of the unseen words with seen words to form the association matrix. The case of adding a new corpus containing new words can be treated similarly. For words unseen in any corpus during training, we also cannot get vectors for them and any rotation matrix is also useless.
>
> > In practice, we can use the MUSE [1] dictionaries you used to evaluate your approach, to train word translation models in a supervised manner.
>
> It has been shown in previous work that supervised translation can work to some extend in the 90s and 00s, but it is only after vectors/pretraining that unsupervised word translation started working in the late 10s. So the community inferred that there is some magic about low-dimensional representations that must be necessary. Thus it is only interesting if we show that the full unsupervised self-learning loop can work in high-dimensions.
>
> In fact, table 1 of your citation [1] shows that supervised vs unsupervised does not make a big difference as long as the self-learning loop gets started unsupervised. Increasingly, as LLMs are shown to perform translation unsupervised, it is increasingly difficult to argue that supervised is always the better choice in practice.
>
> > investigation of downstream tasks, especially for machine translation, is necessary to claim the advantage of coocmap even though it's not conducted in the paper.
>
> The main contribution of coocmap is conceptual by showing that low-dimensions is not necessary and denoising can be effectively performed in high-dimension. Note you can use the same objection against the cited MUSE, vecmap, word2vec etc and the evaluation that you cited. While it would be nice if we can also get results on translating sentences or documents, we thought translating words unsupervised is an interesting enough problem that has a very intuitive evaluation.
>
> Thanks for the questions.
>
> > In the evaluation, why didn't you compare the performance of your coocmap with the reported scores of conventional methods? You can use Semeval 2017 [2] data for this purpose
>
> The included evaluation were deemed enough to support the main points claimed by the paper that you do not need vectors and keeping high dimensions is useful. We do not claim that this method is better than reported scores of "conventional methods" when the amount of data is big enough that everything works.
>
> > Could you expand your approach to using subwords?
> > Furthermore, you need to refer to the fact that current translation models commonly use subwords, not words.
>
> Yes, you can build the subword association matrix as well and even more generally use featurized co-occurences that may include subwords as features. Regardless of how the latest NLP model is tokenized, the word is basic concept recognized by almost everyone, is a better unit for translation than subwords, and used by previous works / evaluations. While subwords represents a good practical trade off point for transformers, it does not render evaluating on words invalid and would only be a complication for this work.
>
> > In the current draft, you use MB to show the size of the data. How did you calculate them? The data you used is compressed or not? Showing the vocabulary size is more helpful for readers to understand.
>
> MB is megabytes of uncompressed text data in utc-8 encoding. The paper clearly states that any data is decompressed if it came compressed. The caption of table 1 also states how to convert the MB to token counts at roughly 0.2 million tokens per MB. It can be counted by various unix and python tools such as 'f.read(1000)' to read 1000 bytes of data. As you pointed out, the token count depends on how you tokenized, thus we picked MB but would not object to token count.
>
> > It's difficult to estimate the computational complexity or space of both coocmap and vecmap by MB of the used data.
>
> In both methods, estimating occurrences or vectors is linear in the size of the data. The remaining matching process is a function of the vocabulary size and independent of the data size. In practice, the data size dependent part of coocmap is very fast, just requiring a counting pass over the data whereas estimating vectors tend to run much slower.

---

> > ### Comment · Reviewer_VTy1 · 2023-08-14
> > **Thank you for answering my questions and comments.**
> >
> > Thank you for answering my questions and comments.
> >
> > >> Not using the rotation matrix means coocmap cannot handle unseen words...
> >
> > > There is no difference in the treatment of unseen words...
> >
> > When using substrings like fastText to represent word embeddings, we can approximately consider vector representations of unseen words. In this case, vecmap can map unseen word embeddings between languages by projecting its rotation matrix, whereas coocmap cannot. Thus, there is a significant difference regarding the treatment of unseen words.
> >
> > - [3] Bojanowski, Piotr, et al. "Enriching Word Vectors with Subword Information." Transactions of the Association for Computational Linguistics 5 (2017): 135-146.
> >
> > >> In practice, we can use the MUSE [1] dictionaries you used to evaluate your approach, to train word translation models in a supervised manner.
> >
> > > It has been shown in previous work that supervised translation...
> >
> > I agree with you regarding the difficulty of arguing that supervised learning is always the better choice in practice. This is because we need to compare performance in supervised and unsupervised learning for each task. We must do that to judge how unsupervised learning is essential for each task if the comparison is possible. The following papers use recent models to compare supervised and unsupervised learning in machine translation. These comparisons indicate the importance of comparing supervised and unsupervised learning for target tasks.
> >
> > - [4] Word alignment (The extended approach for the translation in the 90s to 00s): Dou, Zi-Yi, and Graham Neubig. "Word Alignment by Fine-tuning Embeddings on Parallel Corpora." Proceedings of the 16th Conference of the European Chapter of the Association for Computational Linguistics: Main Volume. 2021.
> > - [5] Machine Translation by LLMs: Zhu, Wenhao, et al. "Multilingual machine translation with large language models: Empirical results and analysis." arXiv preprint arXiv:2304.04675 (2023).
> >
> > >> investigation of downstream tasks, especially for machine translation, is necessary to claim the advantage of coocmap even though it's not conducted in the paper.
> >
> > > The main contribution of coocmap is conceptual by showing that low-dimensions is...
> >
> > I have the same opinion that unsupervised word translation is interesting because of my expectations for low-resource languages. However, the current manuscript does not cover the result in pairs of low-resource languages. Thus, I could not judge coocmap from the practical viewpoint. Regarding the conclusion of the unnecessity of low dimensions, you should consider the performance when training data is large. Figure 2 shows that vecmap-fastext outperforms coocmap in some cases. This result is along with the following theoretical paper that describes the relationship between dataset size and word embedding performance by signal-to-noise ratio.
> >
> > - [6] Yin, Zi, and Yuanyuan Shen. "On the dimensionality of word embedding." Advances in neural information processing systems 31 (2018).
> >
> > > Thanks for the questions.
> >
> > >> In the evaluation, why didn't you compare the performance of your coocmap with the reported scores of conventional methods? You can use Semeval 2017 [2] data for this purpose
> >
> > >The included evaluation were deemed enough to support the main points...
> >
> > Based on the theoretical paper I shared [6], you need to vary both dimension size and data size to check the usefulness of high dimensions.
> >
> > >> Could you expand your approach to using subwords? Furthermore, you need to refer to the fact that current translation models commonly use subwords, not words.
> >
> > > Yes, you can build the subword association matrix as well...
> >
> > Thank you for answering my question with detailed explanations. To show the usefulness of word information on Transformer, you may introduce the case that whole-word masking of BERT can improve performance in downstream tasks. This fact suggests word boundaries are still crucial in Transformer.
> >
> > - [7] Kenton, Jacob Devlin Ming-Wei Chang, and Lee Kristina Toutanova. "Bert: Pre-training of deep bidirectional transformers for language understanding." Proceedings of naacL-HLT. Vol. 1. 2019.
> > - [8] Cui, Yiming, et al. "Pre-training with whole word masking for chinese bert." IEEE/ACM Transactions on Audio, Speech, and Language Processing 29 (2021): 3504-3514.
> >
> > >> In the current draft, you use MB to show the size of the data. How did you calculate them? The data you used is compressed or not? Showing the vocabulary size is more helpful for readers to understand.
> >
> > > MB is megabytes of uncompressed text data in utc-8 encoding...
> >
> > >> It's difficult to estimate the computational complexity or space of both coocmap and vecmap by MB of the used data.
> >
> > > In both methods, estimating occurrences or vectors is linear in the size of the data...
> >
> > Now, I understand that MB in your paper actually corresponds to the vocabulary size for each setting. I appreciate your detailed explanation.

---

> > > ### Author Response · Authors · 2023-08-14
> > >
> > > Thank you for the continued discussions and for the pointers to relevant previous work. We have good answers of your new and old concerns upon clarification. If these address your concerns sufficiently, please consider increasing the score!
> > >
> > > >> Not using the rotation matrix means coocmap cannot handle unseen words...
> > >
> > > > When using substrings like fastText to represent word embeddings... Thus, there is a significant difference regarding the treatment of unseen words.
> > >
> > > We worried/wondered about the effect of the subword information in fasttext as well. Fortunately, we saw no difference at all for this task between fasttext with and without subword information. In the appendix, we noted that "we also checked that the subwords information made no difference by turning them off in fasttext hyperparameters."  The main result of the paper did not need these findings: if the substrings were important, that would only strengthen our baseline where the co-occurrence matrix was more unsupervised than the baseline of fasttext which use more information. So the potential differences is in the direction of favoring the baseline, but actually made no difference at all for this benchmark that do not test on unseen words but may still use substrings to improve representations of seen words.
> > >
> > > Beyond this particular benchmark, co-occurrences can also use subwords if you like. Instead of vector averages, you can count when each substring of word w co-occurred with context c towards w in the style of the cited paper [3] (for example, if "country" is not seen but "count" and "try" were seen, then "country" also gets the co-occurrence of "count" and "try"). You can also use BPE subwords instead of substrings which is more modern.
> > >
> > > > I have the same opinion that unsupervised word translation is interesting because of my expectations for low-resource languages
> > > > However, the current manuscript does not cover the result in pairs of low-resource languages.
> > >
> > > We did cover some low(-ish) resource languages like Finnish and Hungarian, these two with English were reported not to work previously unsupervised by Søgaard et al., 2018, but works with coocmap and improved vecmap and has a very low data requirement of < 100MB of data. While Chinese is not low-resource, we filtered out all Latin characters from its corpus, again measuring the data requirement, showing it to be 100s of MB depending on the domain mismatch. The implication for low-resource are 1) < 100MB is likely fine if domains match otherwise need more 2) the ability to handle mismatched domains is improved with coocmap.
> > >
> > > > [6] Yin, Zi, and Yuanyuan Shen. "On the dimensionality of word embedding." Advances in neural information processing systems 31 (2018).
> > >
> > > > Based on the theoretical paper I shared [6], you need to vary both dimension size and data size to check the usefulness of high dimensions.
> > >
> > >
> > > We did, varying the data was the main experimental setup of the paper! Note the plots are in log of MB of data which goes up to 2000 MB and from 5 to 5000 dimensions. In figure 4 and appendix, we also varied the dimension to show that with coocmap higher dimension is better whereas fasttext has a low optimal dimension. In fact we cited [6] and provide a better explanation for their results in the analysis section. In figure 4, we show that indeed if you train vectors you get an optimal dimension but with the full-dimensional co-occurrence + a sensible matching method, then the higher dimensions the better. It would be interesting to see if this finding applies to their basic word vector evaluation tasks -- actually testing on these monolingual tasks is out of the scope of this paper but I am willing to bet they are wrong.
> > >
> > > > Regarding the conclusion of the unnecessity of low dimensions, you should consider the performance when training data is large.
> > >
> > > We tested on sufficient amount of data (300MB in figure 4) to observe the benefits of higher dimensions. Indeed we only went up to 2GB in figure 3, which you may not consider large. However, [6] argues that lower dimensions is needed with less data (instead more data), so the existing experiments are sufficient to test their hypothesis. Note [6] never varied the data size for their experiments.
> > >
> > > Thanks again for the questions and discussions.

---

> > > > ### Comment · Reviewer_VTy1 · 2023-08-15
> > > > **I appreciate your detailed response.**
> > > >
> > > > I appreciate your detailed response. Of course, I will raise my score if you clear my concerns, and I'm leaning to do it now.
> > > >
> > > > >In the Appendix, we noted that "we also checked that the subwords information made no difference by turning them off in fasttext hyperparameters."
> > > >
> > > > I understand that the decision is based on the preliminary experiments. As a benchmark, subword information may not be an important aspect in contrast to the practical use for downstream tasks requiring to cover unknown words. You can write this point in the Limitation of your paper.
> > > >
> > > > >We did cover some low(-ish) resource languages like Finnish and Hungarian, these two with English were reported not to work previously unsupervised by Søgaard et al., 2018, but works with coocmap and improved vecmap and has a very low data requirement of < 100MB of data.
> > > >
> > > > For the discussion, we need to see the word translation performance for pairs of low(-ish) resource languages, unfortunately not covered in MUSE dictionaries. I believe that you will conduct further experiments on another dataset like XLing-Eval (https://github.com/codogogo/xling-eval) in the final version of your paper.
> > > >
> > > > >We did, varying the data was the main experimental setup of the paper! Note the plots are in log of MB of data which goes up to 2000 MB and from 5 to 5000 dimensions...
> > > >
> > > > In these figures, you did not simultaneously vary both dimensions and data size by comparing vecmap and coocmap. However, taking into account that doing this will make complicated figures, and you can move the figures in Appendix to the main part in the final version, I think the current figures are enough.
> > > >
> > > > >We tested on sufficient amount of data (300MB in figure 4) to observe the benefits of higher dimensions.
> > > >
> > > > Pointing out the uncovered aspects of the conventional theoretical analysis on word embedding is an obvious contribution, especially if you can show its reason, like the difference between continuous and discrete vectors.

---

> > > > > ### Author Response · Authors · 2023-08-16
> > > > >
> > > > > Thanks again for the interesting discussions and valuable suggestions.
> > > > >
> > > > > > As a benchmark, subword information may not be an important aspect in contrast to the practical use for downstream tasks requiring to cover unknown words. You can write this point in the Limitation of your paper.
> > > > >
> > > > > We are happy to elaborate on how subwords may be used to decrease sparsity and to guess at unseen words. Using subwords is indeed less tested in the co-occurrences space and we will include that in limitations.
> > > > >
> > > > > > For the discussion, we need to see the word translation performance for pairs of low(-ish) resource languages, unfortunately not covered in MUSE dictionaries... XLing-Eval (https://github.com/codogogo/xling-eval) ...
> > > > >
> > > > > Thanks for the pointer. Their results show vecmap is already working on all their language pairs using the given vectors. Indeed evaluating on more languages pairs takes more work for us since we are starting with raw data and cannot just download existing vectors to run all pairs. We used English so that we can at least do some sanity check and error analysis. We'd be happy to try some low resource pairs with domain transfer, which is the interesting case for low-resource.
> > > > >
> > > > > > In these figures, you did not simultaneously vary both dimensions and data size by comparing vecmap and coocmap... I think the current figures are enough.
> > > > >
> > > > > Great to hear. Yes, as you noted, we have some figures that varied both dimension and data size in the appendix and even a few more that did not make the appendix as they take a lot of space. We will revisit them for inclusion in the main paper.
> > > > >
> > > > > > Pointing out the uncovered aspects of the conventional theoretical analysis on word embedding is an obvious contribution, especially if you can show its reason, like the difference between continuous and discrete vectors.
> > > > >
> > > > > Yes, this is one of the main contributions which is why the title is "accessing higher dimensions", we show how to use the higher dimensions instead of being limited by incidental properties of dense vectors. If the community finds this interesting, we think further validating this on standard monolingual tasks is a interesting future work.

---

> > > > > > ### Comment · Reviewer_VTy1 · 2023-08-17
> > > > > > **You've cleared my concerns.**
> > > > > >
> > > > > > I'm thankful to you for your detailed responses. You've answered all my questions, and that has cleared my concerns. I'll raise my recommendation score.

---

### Author Rebuttal · Authors · 2023-08-10

We thank all reviewers for the insightful and helpful reviews. We will aim to improve the draft using the feedback and make it more clear. A few common points are emphasized here and other points and questions are addressed in individual responses.

1. We are the first to use just co-occurrences to achieve fully unsupervised word translation. To do this, we needed the normalization and relative measurements which was developed for vector methods but turned out to be even more crucial for co-occurrence methods. Interestingly, it turns out that these methods rather than vectors were the key. In addition to normalization and relative measurements, which enabled co-occurences to work at all but slightly less accurate than vectors, clip and drop was needed to really make higher dimensional vectors more robust and more accurate than well-trained low-dimensional vectors on exactly the same data.

2. The main conceptual contribution is to change our understanding from "dense vectors work better in every NLP task than sparse vectors" (Jurafsky and Martin, 2023,6.8) to "there is useful information in higher dimensions" accessible after applying simple techniques such as normalization, relative measurement plus drop and clip. We back up this new view with detailed analysis and experiments on dimensions, which is in retrospect more plausible/intuitive than the prevailing view of the community.

On the other hand, we do not consider achieving slightly higher accuracy as a main contribution nor very interesting. In fact, figure 2 shows that vecmap often has slightly better accuracy than the basic coocmap with enough data. We do not think the word translation task is sensitive enough such that gaining a few percentage more accuracy is important/interesting. We are aware of its limitations both pointed out by some reviewers and by previous work (identical words, lots of proper nouns). On the other hand, we are still happy to use word translation as the evaluation because it is very intuitive and it is absolutely good enough for testing if unsupervised translation is succeeding at all. These known issues are not relevant if we only focus on the binary phenomenal and data efficiency instead of a few percentage difference in the accuracy.

---

### Decision · Program_Chairs · 2023-09-21

**Decision:**

Accept (poster)

**Comment:**

The paper proposes a method that uses high-dimensional co-occurrence counts or their lower-dimensional approximations for unsupervised machine translation. The evaluation results showed that the proposed approach outperformed conventional approaches. The paper can be further strengthened with the inclusion of better motivation and clarity in the description of the experimental setup.